# RAB13 mRNA compartmentalisation spatially orients tissue morphogenesis

Guilherme Costa[1,2,*] , Joshua J Bradbury[1], Nawseen Tarannum[1] & Shane P Herbert[1,**]

## Abstract

Polarised targeting of diverse mRNAs to cellular protrusions is a hallmark of cell migration. Although a widespread phenomenon, definitive functions for endogenous targeted mRNAs and their relevance to modulation of *in vivo* tissue dynamics remain elusive. Here, using single-molecule analysis, gene editing and zebrafish live-cell imaging, we report that mRNA polarisation acts as a molecular compass that orients motile cell polarity and spatially directs tissue movement. Clustering of protrusion-derived RNAseq datasets defined a core 192-nt localisation element underpinning precise mRNA targeting to sites of filopodia formation. Such targeting of the small GTPase RAB13 generated tight spatial coupling of mRNA localisation, translation and protein activity, achieving precise subcellular compartmentalisation of RAB13 protein function to create a polarised domain of filopodia extension. Consequently, genomic excision of this localisation element and perturbation of *RAB13* mRNA targeting—but not translation—depolarised filopodia dynamics in motile endothelial cells and induced mispatterning of blood vessels in zebrafish. Hence, mRNA polarisation, not expression, is the primary determinant of the site of RAB13 action, preventing ectopic functionality at inappropriate subcellular loci and orienting tissue morphogenesis.

**Keywords** angiogenesis; endothelial cell; filopodia; mRNA targeting; zebrafish
**Subject Categories** Development; Membranes & Trafficking; Translation & Protein Quality
**The EMBO Journal (2020) 39: e106003**

## Introduction

Dynamic subcellular polarisation of a myriad of proteins fundamentally shapes the front-rear orientation and directed movement of motile cells during tissue formation (reviewed in Mayor & Etienne-Manneville, 2016). In parallel, cell migration is associated with subcellular polarisation of numerous mRNAs (reviewed in Herbert & Costa, 2019), but whether this phenomenon is also relevant to the modulation of tissue dynamics remains an open question. However, in many other biological contexts, mRNA localisation and local translation are well-established as key determinants of cell polarity (Buxbaum *et al*, 2015). This mode of spatial control of gene expression contributes to polarised cellular responses in broad contexts, ranging from axon growth (Leung *et al*, 2006; Yao *et al*, 2006) and synaptic function (e.g. Kang & Schuman, 1996; Lyles *et al*, 2006; Younts *et al*, 2016) to epithelial polarity (Nagaoka *et al*, 2012; Moor *et al*, 2017). Moreover, there is a wealth of data in diverse cell types demonstrating that large numbers of mRNAs are co-distributed together at distinct subcellular sites, which has led to the idea that such mRNA polarisation functions to generate local transcriptomes (reviewed in Engel *et al*, 2020). This suggests that clusters of mRNAs encoding proteins belonging to common complexes and biological pathways co-localise to participate in local processes (e.g. Mingle *et al*, 2005; Hotz & Nelson, 2017). Indeed, it has been proposed that such co-distribution of mRNAs also ensures the fidelity of interactions between locally produced proteins in rapidly changing cell environments (Weatheritt *et al*, 2014). Nevertheless, during cell migration, the impact of mRNA polarisation on the control of translated protein function, local assembly of the migratory machinery and motile cell polarity remain poorly understood, as does the *in vivo* relevance of this phenomenon.

The polarised localisation of mRNAs is driven by *cis*-localisation elements (LEs) that are commonly found in 3′ untranslated regions (UTRs) (Andreassi & Riccio, 2009; Mayr, 2016). Indeed, alternative 3′UTRs have been shown to control the spatial localisation of mRNAs (Taliaferro *et al*, 2016; Tushev *et al*, 2018) and modulate protein distribution (An *et al*, 2008; Ciolli Mattioli *et al*, 2019), suggesting that tight control of LE usage may underpin spatial regulation of gene expression. Although the complex sequence and structural composition of LEs render the identification of conserved RNA motifs within large groups of co-localised mRNAs a challenging task, several individual LEs have been characterised in detail using diverse model organisms (reviewed in Jambhekar & Derisi, 2007). In vertebrate cells, LEs ranging from just a few to hundreds of nucleotides in length have been identified within 3′UTRs (e.g. Mowry & Melton, 1992; Kislauskis *et al*, 1994; Ainger *et al*, 1997). However, considering the difficulties in manipulating endogenous transcripts, our understanding of LE function during complex tissue formation in vertebrate organisms remains relatively poor.

1 Faculty of Biology, Medicine and Health, University of Manchester, Manchester, UK
2 Wellcome-Wolfson Institute for Experimental Medicine, Queen's University of Belfast, Belfast, UK
*Corresponding author. Tel: +44 0289 097 6020; E-mail: g.costa@qub.ac.uk
**Corresponding author. Tel: +44 0161 275 1140; E-mail: shane.herbert@manchester.ac.uk

Here, using novel reporter transgenics in zebrafish embryos and targeted gene editing, we shed light on the function of LE-mediated mRNA polarisation in the control of tissue formation *in vivo*. Following clustering analysis of transcriptome-wide data, we define a core group of 5 mRNAs that are universally targeted to the leading edge of migratory cells *in vitro*. Moreover, we identify a conserved RNA motif within the 3′UTRs of these genes and a 192-nt LE containing four of these motifs that is sufficient to target transcripts to polarised sites of filopodia remodelling. Excision of this LE in transcripts encoding the small GTPase RAB13 perturbs mRNA localisation (but not translation) and was sufficient to depolarise RAB13-mediated filopodia dynamics in motile endothelial cells *in vitro*. Hence, *RAB13* mRNA polarisation achieves precise spatial compartmentalisation of RAB13 protein activity to the front of migrating cells and blocks ectopic protein action at inappropriate subcellular loci. Consequently, excision of the *rab13* LE in zebrafish embryos also perturbed mRNA polarisation and induced mispatterning of nascent blood vessels. Altogether, our findings show that mRNA polarisation spatially restricts protein activity to precisely orient motile cell polarity and tissue movement *in vivo*.

# Results

## Clustering of RNAseq datasets identifies mRNAs exhibiting universal targeting to protrusions

To explore the ability of targeted mRNAs to direct tissue formation, we first aimed to define mRNA localisation motifs driving transcript polarisation in motile endothelial cells (ECs), as an initial step towards probing their function in coordinating blood vessel morphogenesis *in vivo*. As a starting point, we identified 233 transcripts enriched in fractionated cellular protrusions of migrating primary human umbilical vein ECs (HUVECs) *in vitro* (Fig 1A and B; Table EV1). ECs were seeded on Transwells in low serum and induced to migrate upon addition of VEGF-A to the lower chamber (Fig 1A). Consequently, the motile protrusions and trailing cell bodies of ECs were separated and protrusion-enriched transcripts identified by RNAseq (Fig 1B; Table EV1). *k*-means clustering analysis of these data alongside RNAseq datasets from unrelated cell types (NIH/3T3 fibroblasts (Wang *et al*, 2017), MDA-MB231

metastatic breast cancer cells (Mardakheh *et al*, 2015), induced neuronal cells (Zappulo *et al*, 2017)) revealed unexpected cell type-specific diversity to transcript polarisation, with only five mRNAs exhibiting universal targeting to protrusions in all cell types tested (cluster *k* 5; *RAB13*, *TRAK2*, *RASSF3*, *NET1*, *KIF1C*; Figs 1B and C, and EV1A). Strikingly, cluster *k* 5 mRNAs shared near-identical spatial distributions by single-molecule FISH (smFISH) (Raj *et al*, 2008; Tsanov *et al*, 2016), being highly polarised to cellular protrusions relative to a control transcript, *GAPDH* (Fig 1D–G). Indeed, all cluster *k* 5 mRNAs exhibited a significantly higher Polarisation Index (PI) than *GAPDH* when co-detected in migrating ECs (Fig 1E) and the ratio of *k* 5 mRNA PI to *GAPDH* PI was consistently greater than one in individual cells (Fig 1F). Moreover, *k* 5 transcripts were highly spatially distinct from other clusters, such as mRNAs of cluster *k* 7 that exhibited less-polarised perinuclear targeting (Figs 1G and H, and EV1B and D). Likewise, protrusion localisation of the well-established polarised mRNA *ACTB* (Condeelis & Singer, 2005) was significantly more diffuse than *k* 5 mRNAs, as were other cluster *k* 2 members such as *PPDPF* (Figs 1G and I, and EV1B and C). Finally, protrusion-enriched mRNAs were also tightly clustered according to protein function (Table EV2), with *k* 5 transcripts specifically encoding cell periphery-associated modulators of vesicle trafficking and membrane remodelling (Tommasi *et al*, 2002; Brickley *et al*, 2005; Kopp *et al*, 2006; Srougi & Burridge, 2011; Wu *et al*, 2011). Hence, tight coupling of distinct mRNA spatial distributions with discrete protein functionalities likely indicates that the universal polarisation of *k* 5 mRNAs is a key functional requirement in processes common to all motile cells.

## Clustering of RNAseq datasets defines an RNA motif enriched in 3′UTR sequences that target *k* 5 mRNAs to protrusions

Considering that the polarisation of cluster *k* 5 mRNAs was particularly acute, highly stereotyped and uniquely conserved amongst cell types (Fig 1C–G), we hypothesised that these transcripts employed common targeting mechanisms. Indeed, using the MEME Suite (Bailey & Elkan, 1994) we detected consistent repeat use of a conserved sequence motif in the 3′UTRs of all *k* 5 transcripts (Fig 2A and B, and Table EV3). This motif distinguished the *k* 5 mRNAs from other identified mRNA clusters, which contained the motif at much lower frequency (Fig 2C). Moreover, this motif was

---

**Figure 1. Clustering of RNAseq datasets identifies mRNAs exhibiting universal targeting to protrusions in diverse cell types.**

A   Strategy used to screen for mRNAs enriched in motile protrusions of HUVECs migrating through Transwell membranes.

B   RNAseq data are plotted in $\log_2$ fold change (FC) levels of protrusions over cell bodies against adjusted $-\log_{10}$ false discovery rate (FDR) ($n$ = 2 replicates, average FC values are represented). The horizontal dashed line marks the FDR ($q$) = 0.05 threshold; vertical dashed lines mark the FC = 0.625 (left) or FC = 1.6 (right) thresholds.

C   Heat map represents the *k*-means clustering of transcript $\log_2$ FC levels (protrusions over cell bodies) extracted from RNAseq datasets published elsewhere. The corresponding HUVEC FC levels are shown in parallel.

D   smFISH co-detection of *k* 5 mRNAs and *GAPDH* in subconfluent motile HUVECs.

E   Polarisation Index (PI) of *k* 5 mRNAs and *GAPDH* in co-detected in HUVECs ($n \geq 28$ cells; **$P < 0.01$, ***$P < 0.001$, ****$P < 0.0001$; Wilcoxon test).

F   *k* 5 mRNA PIs plotted against respective *GAPDH* PIs. The slope of the coloured lines represents the average *k* 5 mRNA/*GAPDH* PI ratio; the dashed grey line represents a 1:1 ratio ($n \geq 28$ cells).

G   Top: distribution pattern of mRNAs clustered in *k* 2, *k* 5, *k* 7 and *GAPDH*. Bottom: smFISH co-detection of exemplar *k* 7/*k* 2 mRNAs and *GAPDH* in subconfluent motile HUVECs.

H   PIs of *k* 7 mRNAs and *GAPDH* co-detected in HUVECs ($n \geq 25$ cells; *$P < 0.05$, **$P < 0.01$, ***$P < 0.001$; paired *t* test).

I   PIs of *k* 2 mRNAs and *GAPDH* co-detected in HUVECs ($n \geq 19$ cells; *$P < 0.05$; Wilcoxon test).

Data information: arrows indicate orientation of RNA localisation; yellow dashed lines outline cell borders; red circles highlight smFISH spots; scale bars = 20 μm (D, G). Bar charts are presented as means ± s.d.

---

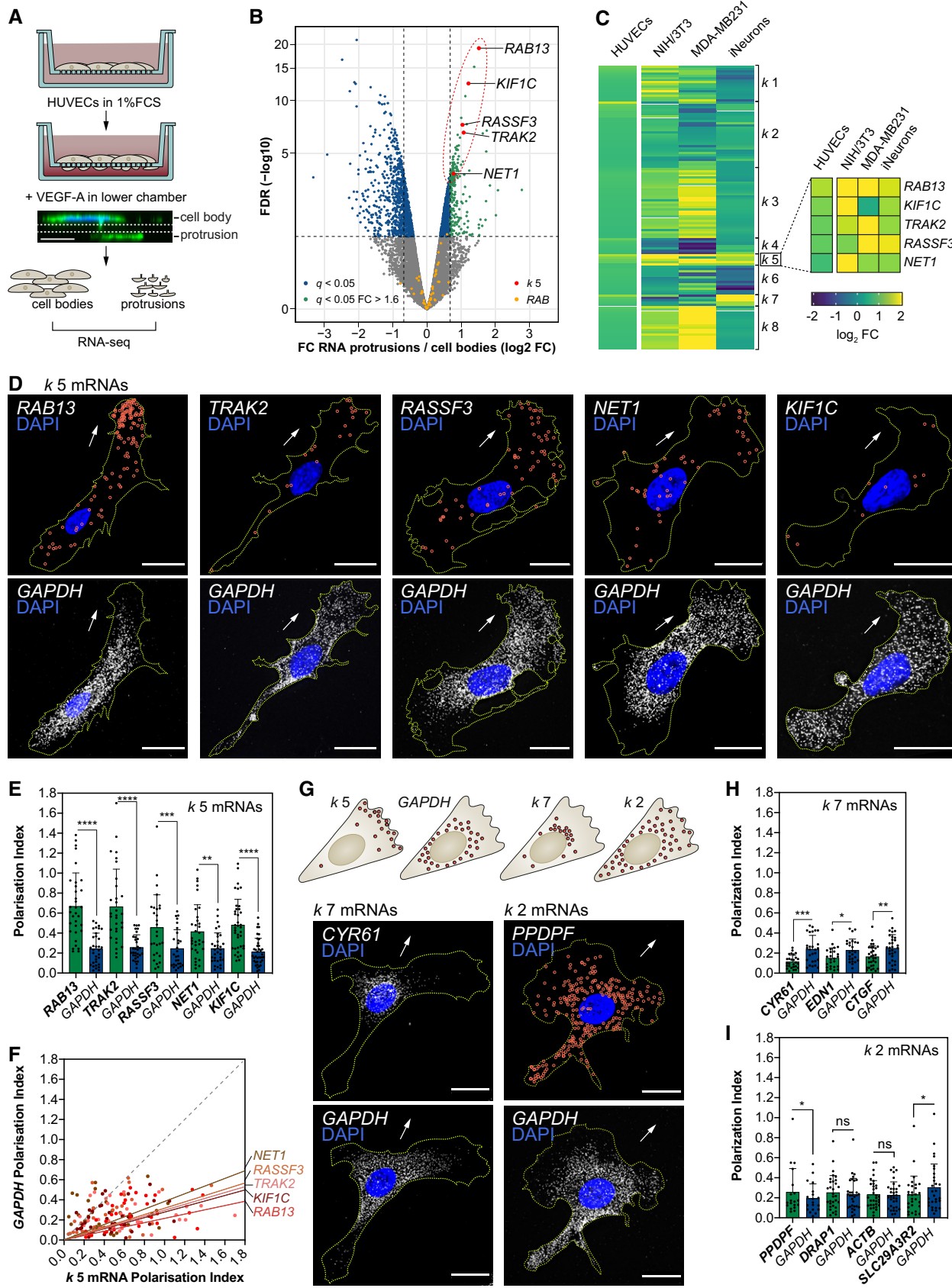

**Figure 1.**

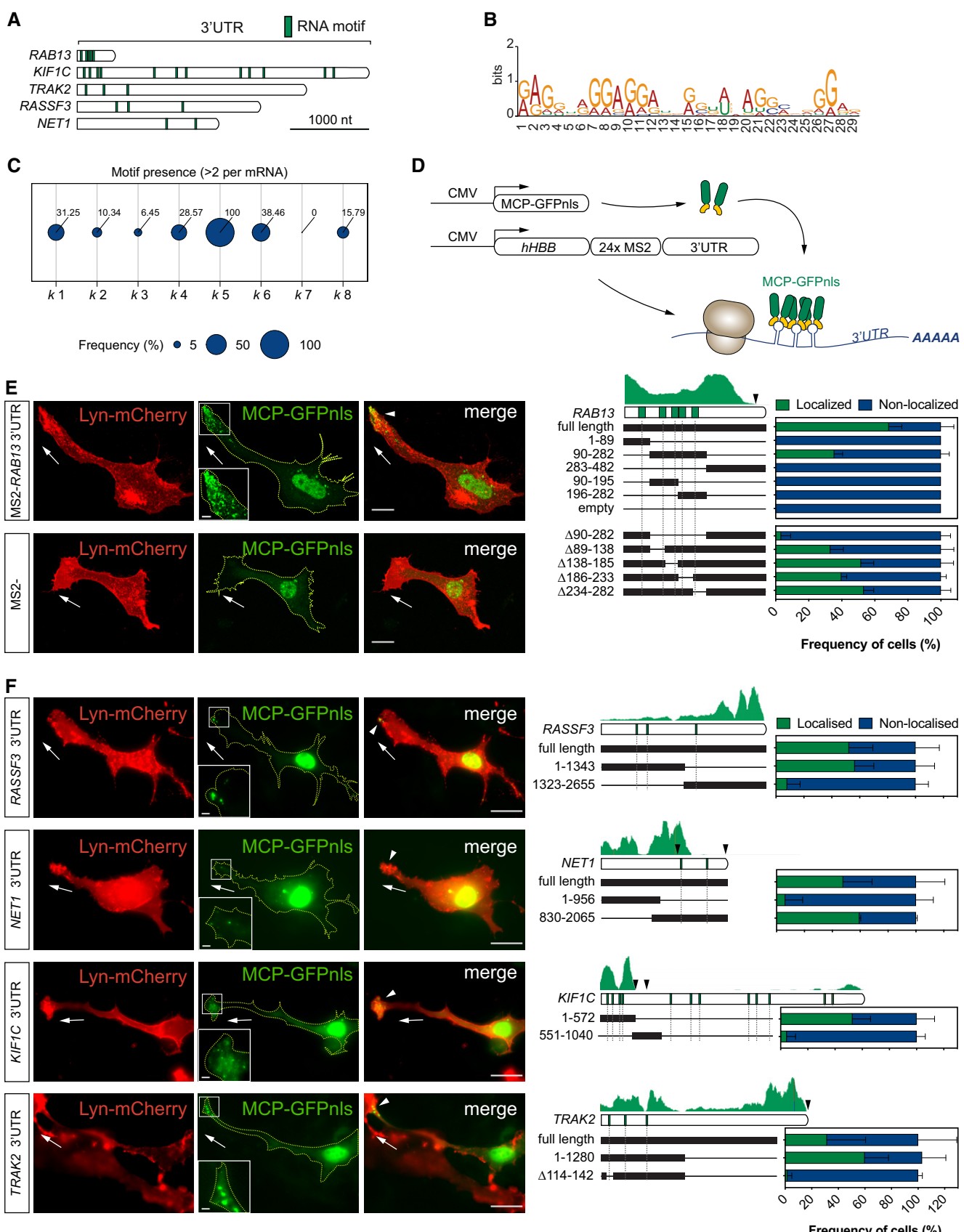

**Figure 2.**

Figure 2. Clustering of RNAseq datasets defines an RNA motif enriched in 3′UTR sequences that target *k* 5 mRNAs to protrusions.

A   Diagram of *k* 5 mRNA 3′UTRs and relative positions of the RNA motif shared between transcripts.
B   RNA motif over-represented in *k* 5 mRNA 3′UTRs.
C   Frequency of mRNAs within each *k*-means cluster containing at least 2 of the RNA motif over-represented in *k* 5 mRNAs.
D   Scheme depicts the *in vitro* MS2 system strategy. CMV promoter-driven expression of MCP-GFPnls and *hHBB*-24xMS2-tagged *RAB13* 3′UTR. The visualisation of MCP-GFPnls bound to 24xMS2 allows the identification of the minimal region in the 3′UTR of *RAB13* necessary for its localisation.
E   Left: representative subconfluent motile cells co-transfected with plasmids expressing Lyn-mCherry, MCP-GFPnls and 24xMS2-*RAB13* 3′UTR or 24xMS2. Right: percentage of cells with MCP-GFPnls localised to protrusions when co-transfected with full length or deletion versions of *RAB13* 3′UTR (*n* ≥ 3 experiments).
F   Left: representative cells co-transfected with plasmids expressing Lyn-mCherry, MCP-GFPnls and 24xMS2-*k* 5 3′UTRs. Right: percentage of cells with MCP-GFPnls localised to protrusions when co-transfected with full length or deletion versions of *k* 5 3′UTR (*n* ≥ 3 experiments).

Data information: white arrowheads indicate non-nuclear localisation of MCP-GFPnls; arrows indicate the orientation of RNA localisation; yellow dashed lines outline cell borders; scale bars = 20 μm (E, F); scale bars in insets = 5 μm (E) and 2 μm (F). For each *k* 5 mRNA 3′UTR, a diagram of the full-length 3′UTR and the positions of the RNA motif is shown together with the respective RNAseq mapped reads from HUVEC protrusions; black arrowheads indicate predicted polyadenylation sites (E, F). Bar charts are presented as means ± s.d.

striking in its clustering as five repeats within a short 3′UTR region of *RAB13*, a known polarised mRNA (Mili *et al*, 2008; Jakobsen *et al*, 2013; Moissoglu *et al*, 2019) (Fig 2A). To interrogate its function, we tagged the non-localising human *HBB* coding sequence with both the *RAB13* 3′UTR and the reporter MS2 hairpin repeats (Bertrand *et al*, 1998; Mili *et al*, 2008; Fig 2D). Following co-expression with the MS2 capping protein (MCP)-GFPnls that is usually confined to the nucleus (Fusco *et al*, 2003), the localisation of MS2-tagged mRNAs can be monitored through changes to the spatial distribution of the GFP signal (Fig 2D). Potent localisation properties of the motif-containing *RAB13* 3′UTR were confirmed upon expression of the MS2 reporter system in ECs (Fig 2E). Using this approach in combination with truncations or deletions of the *RAB13* 3′UTR, we identified a minimal 192-nt LE encompassing four motif repeats that was both necessary and sufficient to exclusively polarise mRNA at motile EC protrusions (region 90–282 in Fig 2E). Furthermore, similar truncation and deletion analysis of the remaining *k* 5 mRNAs confirmed that these transcripts employ common *cis*-regulatory mechanisms, as mRNA targeting ability was consistently reliant on motif-containing 3′UTR regions (Fig 2F). Indeed, precise deletion of a single motif in *TRAK2* was sufficient to entirely block mRNA targeting (Fig 2F). Of note, unlike *TRAK2*, more than two functional motifs were required to drive *RAB13* polarisation, as deletions of individual motifs were tolerated and constructs containing two motifs were insufficient to

drive *RAB13* mRNA localisation (Fig 2E). Hence, distinct cluster *k* 5 mRNAs may exhibit different minimal requirements for the number of motifs needed to drive targeting.

**CRISPR-Cas9 excision of the 3′UTR localisation element of *RAB13* disrupts mRNA targeting**

As *RAB13* was the only identified RAB small GTPase to exhibit such mRNA polarisation (Fig 1B), we hypothesised that the identified LE and targeting of the transcript were critical for RAB13 protein function in motile cells. However, studies probing the precise function of endogenous polarised mRNAs in motile cells are currently lacking, predominantly due to difficulties identifying targeting motifs and the potential propensity for genomic manipulation to perturb transcript stability and/or translation. However, precise genomic excision of the *RAB13* minimal LE in ECs using CRISPR-Cas9 tools did not perturb *RAB13* mRNA or protein expression (Fig 3A–G). ECs were transfected with both a GFP-expressing plasmid and CRISPR-Cas9 ribonucleoprotein complexes targeting the 90–282-nt LE in exon 8 of *RAB13*, prior to expansion of GFP-expressing clones (Fig 3A). Clones were then selected for either biallelic deletion of the LE (ΔLE) or presence of the full-length *RAB13* 3′UTR (Wt) (Fig 3B) and were sequenced to confirm specific deletion of the LE in mutated clones (Figs 3C and EV2A). Moreover, RNAseq analysis verified that overall splicing of *RAB13*

Figure 3. CRISPR-Cas9 excision of the 3′UTR localisation element of *RAB13* disrupts mRNA targeting.

A   CRISPR-Cas9 strategy to derive HUVECs with an excision of the LE in the *RAB13* 3′UTR (ΔLE) and parallel generation of wild-type (Wt) control cells. The Wt *RAB13* exon 8 is represented with its coding sequence in dark and the 3′UTR in clear boxes. The 5′ and 3′ gRNA-targeted regions are represented with green lines. Arrows: relative positions of the forward (*f*) and reverse (*r*) PCR primers used to identify HUVECs with CRISPR-Cas9-mediated excision of the LE.
B   Representative genotyping PCR demonstrates the band size shift in ΔLE HUVECs.
C   Detailed DNA sequence depicting nucleotide positions within the *RAB13* 3′UTR of Wt and ΔLE HUVECs.
D   Wt and ΔLE HUVEC RNAseq mapped reads depicting *RAB13* exon usage.
E   Quantification of *RAB13* mRNA smFISH spot number in Wt and ΔLE HUVECs (*n* = 3 experiments; ns: not significant; unpaired *t* test).
F   Number of *RAB13* mRNA smFISH spots plotted against the respective Polarisation Index (PI) (*n* = 29 cells; ns: not significant; linear regression).
G   Left: representative Western blotting (WB) of Wt and ΔLE HUVECs. Right: densitometry analysis of WB data (*n* = 3 samples; ns: not significant; unpaired *t* test).
H   smFISH co-detection of *RAB13* and control *GAPDH* in Wt and ΔLE motile HUVECs cultured under subconfluent conditions.
I   PI of *RAB13* and *GAPDH* co-detected in Wt and ΔLE HUVECs (*n* = 29 cells; ***P < 0.001, ns: not significant; Mann–Whitney test).
J   *RAB13* PI plotted against respective *GAPDH* PI. The slope of the coloured lines represents the average *RAB13*/*GAPDH* PI ratio; the dashed grey line represents a 1:1 ratio (*n* = 29 cells).

Data information: 3 Wt and 3 ΔLE HUVECs independent clones were used to collect data (E–J). Arrows indicate orientation of RNA localisation; yellow dashed lines outline cell borders; red circles highlight smFISH spots; scale bars = 20 μm (H). Bar charts are presented as means ± s.d.
Source data are available online for this figure.

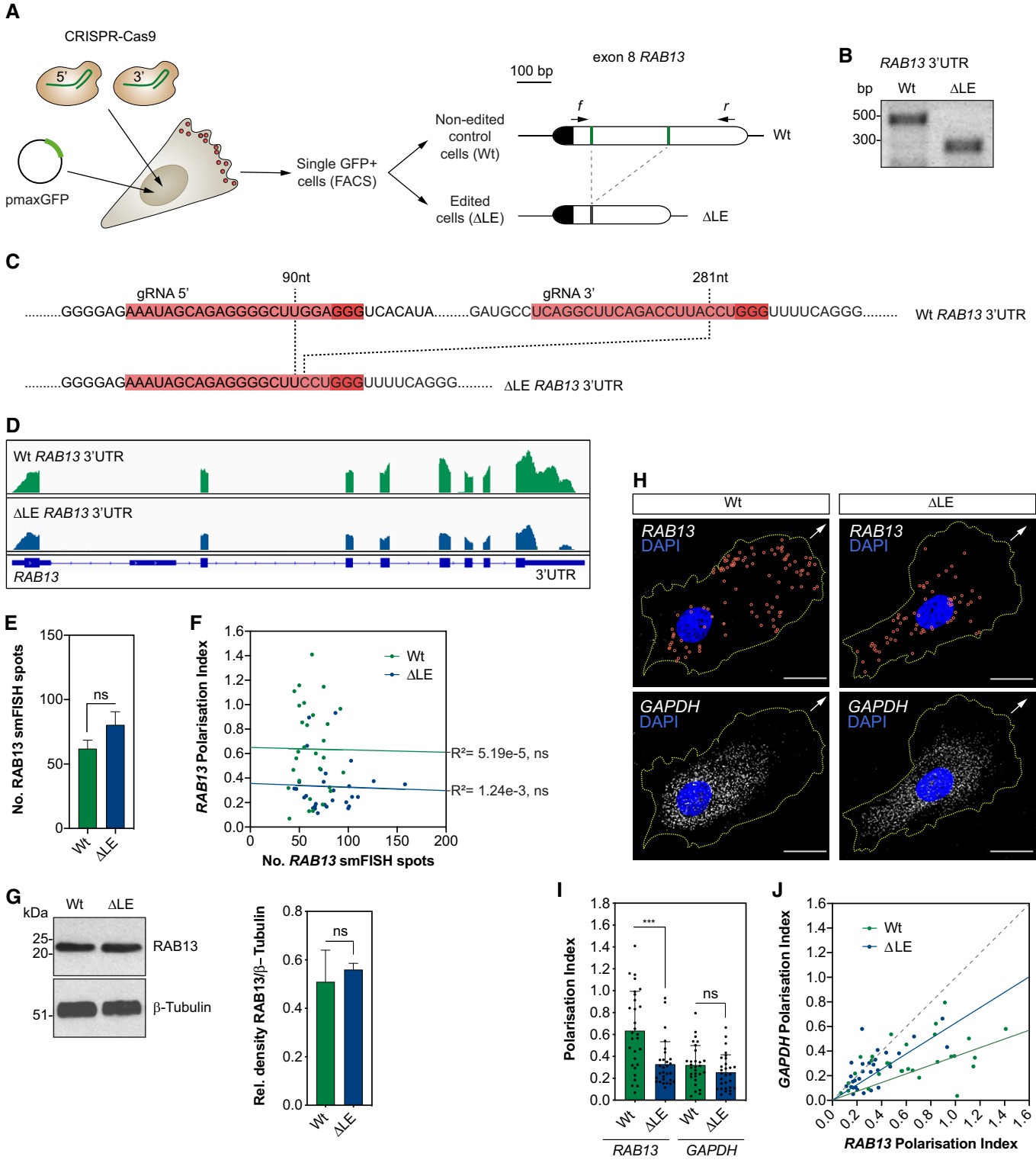

Figure 3.

mRNA was unaffected by genomic excision of the LE (Fig 3D) and confirmed the high specificity of these CRISPR-Cas9 tools, as no nucleotide mismatches were observed at any putative low-frequency off-target sites (Fig EV2B). Importantly, excision of the LE did not perturb *RAB13* mRNA levels or protein expression

(Fig 3E–G), but did eradicate the polarised spatial pattern of *RAB13* localisation, such that the transcript became diffusely distributed in ECs similar to *GAPDH* (Fig 3H). In particular, quantification of the PI of co-detected *RAB13* and *GAPDH* mRNAs revealed that loss of the LE consistently reduced *RAB13*

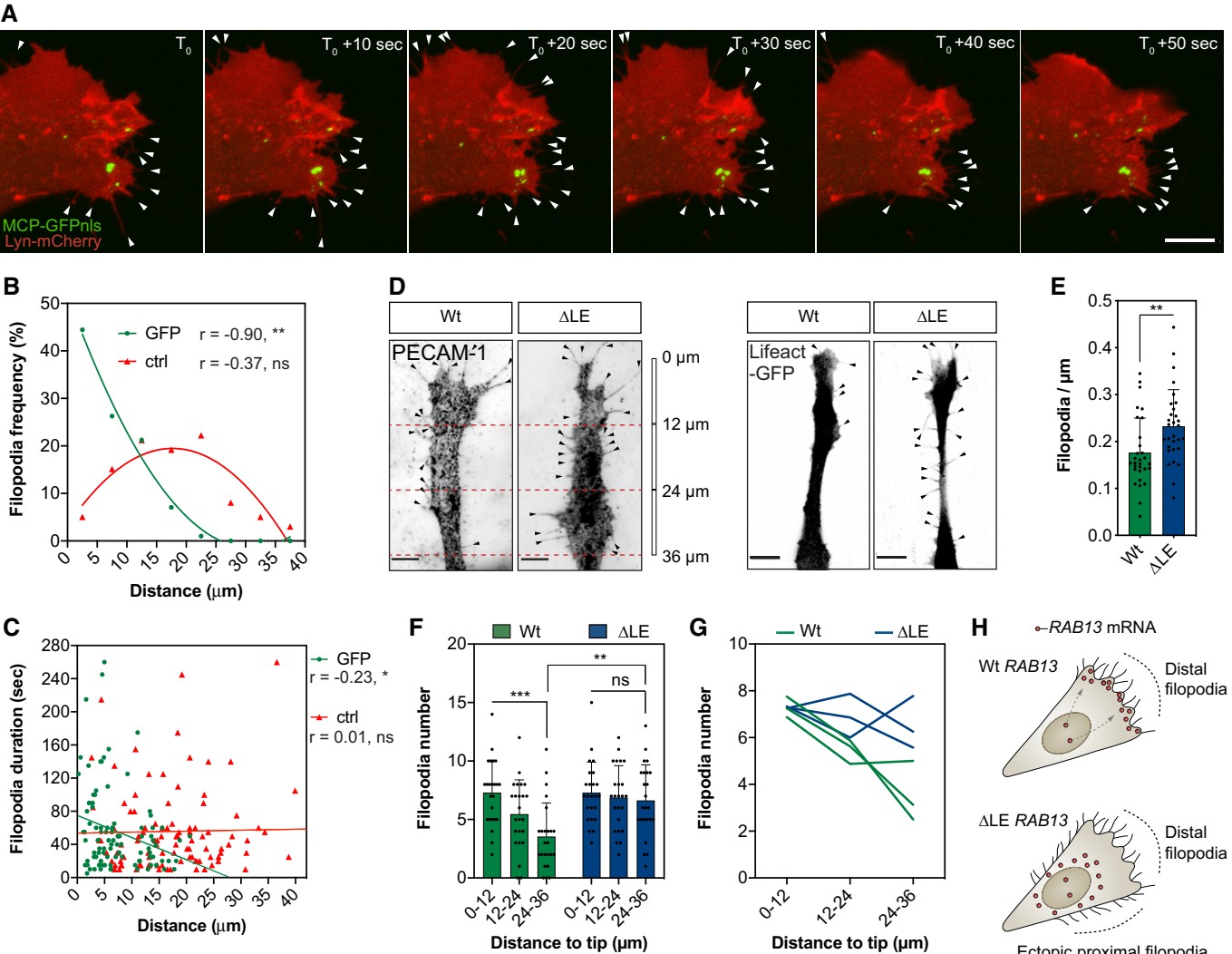

**Figure 4. *RAB13* mRNA polarisation spatially orients filopodia dynamics.**

A   Representative time-lapse microscopy of a bEnd.3 cell co-transfected with plasmids expressing Lyn-mCherry, MCP-GFPnls and 24xMS2-*RAB13* 3′UTR.
B   Frequency of newly formed filopodia formed within 5-μm intervals relative to the nearest MCP-GFPnls particle or a randomised (ctrl) position (*n* = 99 filopodia; **P < 0.01, ns: not significant; Pearson's *r* correlation).
C   Distance of newly formed filopodia to MCP-GFPnls or a ctrl position plotted against filopodia duration (*n* = 99 filopodia; *P < 0.05, ns: not significant; Spearman's *r* correlation).
D   Wt and ΔLE HUVECs co-cultured on fibroblast monolayers. Endothelial cells were identified either with an antibody against the endothelial cell marker PECAM-1 (left) or through expression of a nucleofected plasmid encoding the cytoskeletal marker Lifeact-GFP (right).
E   Number of filopodia detected in co-cultured HUVECs (*n* = 30 cells; **P < 0.01; unpaired *t* test).
F   Number of filopodia detected in co-cultured HUVECs within 12-μm intervals relative to cell distal tip (*n* = 30 cells; **P < 0.01, ***P < 0.001, ns: not significant; one-way ANOVA with Bonferroni's correction).
G   Number of filopodia detected in individual clones of co-cultured HUVECs within 12-μm intervals relative to cell distal tip.
H   Illustration of the spatial relationship between *RAB13* mRNA localisation and sites of filopodia production.

Data information: 3 Wt and 3 ΔLE HUVECs independent clones were used to collect data (D–G). Arrowheads indicate filopodia (A, D); scale bars = 10 μm (A) and 6 μm (D). Bar charts are presented as means ± s.d.

polarisation to levels equivalent to *GAPDH* controls (Fig 3I and J). Finally, correlation of *RAB13* mRNA spot count versus *RAB13* mRNA PI revealed that mRNA polarisation is actually independent of total mRNA levels and further confirmed that it is perturbed upon excision of the LE (Fig 3F). Hence, genomic excision of the 3′UTR LE specifically perturbs *RAB13* mRNA polarisation.

## *RAB13* mRNA polarisation spatially orients filopodia dynamics

RAB13 is an established modulator of cortical F-actin crosslinking and cytoskeletal remodelling at leading front of migrating cells, via interaction with its effector protein, MICAL-L2 (Sakane *et al*, 2012, 2013; Ioannou *et al*, 2015). Consistent with this function, live-cell

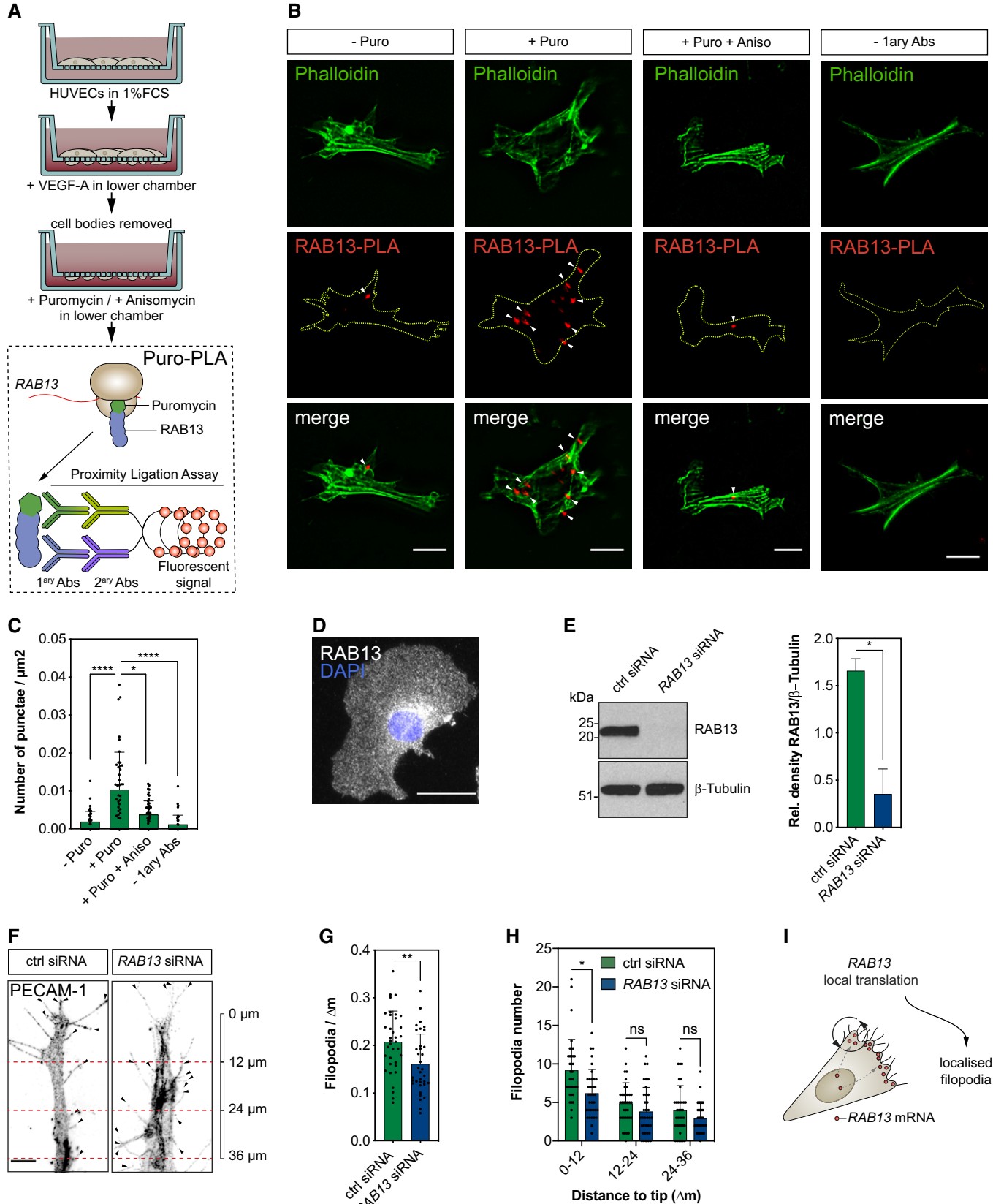

**Figure 5.**

◀

**Figure 5. mRNA polarisation achieves spatial compartmentalisation of RAB13 translation and protein function.**

A   Strategy used to detect local protein synthesis in protrusions formed by HUVECs migrating through Transwell membranes.
B   Representative Puro-PLA experiments detecting newly synthesised RAB13 in HUVEC protrusions present in the lower side of Transwell membranes. Puro: puromycin; Aniso: anisomycin; 1ary Abs: primary antibodies.
C   Quantification of RAB13 Puro-PLA punctae normalised to protrusion area ($n \geq 40$ protrusions; *$P < 0.05$, ****$P < 0.0001$; Kruskal–Wallis test with Dunn's correction).
D   Representative RAB13 IF assay on migrating HUVECs.
E   Left: representative Western blotting (WB) of siRNA-transfected HUVECs. Right: densitometry analysis of WB data ($n = 3$ samples; *$P < 0.05$; paired $t$ test).
F   Control (ctrl) and *RAB13* siRNA-treated HUVECs co-cultured on fibroblast monolayers. Endothelial cells were identified with an antibody against the endothelial cell marker PECAM-1.
G   Number of filopodia detected in co-cultured HUVECs ($n \geq 35$ cells; **$P < 0.01$; unpaired $t$ test).
H   Number of filopodia detected in co-cultured HUVECs within 12-μm intervals relative to cell distal tip ($n \geq 35$ cells; *$P < 0.05$, ns: not significant; Kruskal–Wallis test with Dunn's correction).
I   Illustration of the spatial relationship between the sites of *RAB13* mRNA localisation, local translation and RAB13 protein-mediated filopodia distribution.

Data information: white arrowheads indicate Puro-PLA punctate; yellow dashed lines outline protrusion borders (B); black arrowheads indicate filopodia (F); scale bars = 10 μm (B, D) and 6 μm (F). Bar charts are presented as means ± s.d.
Source data are available online for this figure.

imaging of *RAB13* 3′UTR dynamics employing MS2 reporter constructs revealed enriched targeting of mRNA to sites of incipient filopodia formation in cell protrusions, suggesting a tight spatial coupling between *RAB13* mRNA localisation and RAB13 protein activity (Fig 4A–C and Movie EV1). Furthermore, quantification revealed that filopodia preferentially emerged in close proximity to GFP particles transported by the *RAB13* 3′UTR reporter, but not with random cytoplasmic spot positions (Fig 4B). In addition, there was a putative causal relationship between *RAB13* mRNA proximity and increased filopodia stability (Fig 4C). Likewise, induction of cell migration drove a significant increase in the levels and polarisation of *RAB13* mRNA *in vitro* (Fig EV3), further indicating a dynamic involvement in leading-edge remodelling and establishment of cell polarity. Hence, these data revealed that polarisation of *RAB13* mRNA may spatially compartmentalise RAB13-mediated F-actin remodelling to orient motile cell polarity. Consistent with this hypothesis, loss of *RAB13* mRNA polarisation—but not loss of expression—was indeed sufficient to depolarise filopodia dynamics in motile ECs (Fig 4D–G). When co-cultured with fibroblasts to mimic polarised blood vessel sprouting (Hetheridge *et al*, 2011), Wt ECs exhibited highly polarised filopodia extensions biased towards the leading edge of motile cells (Fig 4D and F). In contrast, filopodia in ECs lacking the *RAB13* LE (ΔLE) were no longer spatially compartmentalised and became ectopically homogenously distributed along the distal–proximal cell axis (Fig 4D and F). Importantly, these observations were consistent between individual Wt and ΔLE CRISPR-Cas9 clones (Fig 4G). Consequently, mutant ECs exhibited a significant increase in overall filopodia frequency (Fig 4E). Hence, these data indicate that tight control of *RAB13* mRNA localisation spatially specifies a polarised domain of filopodia extension in motile cells (Fig 4H).

## mRNA polarisation achieves spatial compartmentalisation of RAB13 translation and protein function

These striking observations suggested that *RAB13* mRNA polarisation acts to exclusively spatially compartmentalise RAB13-mediated filopodia extension at distal sites. As such, targeting of *RAB13* mRNA and local translation could effectively block ectopic protein function at inappropriate subcellular loci to orient motile cells. However, this could only be achieved if the sites of *RAB13* mRNA localisation, translation and protein function were all

tightly spatially coupled. Indeed, such coupling may be consistent with long-standing proposals that newly translated RABs form a discrete protein pool from mature RABs, potentially with distinct interaction partners (e.g. specific RAB escorting proteins and GDP dissociation inhibitors) and separate biological functions (Pfeffer *et al*, 1995; Shen & Seabra, 1996; Seabra *et al*, 2002). Hence, local translation of polarised *RAB13* transcript may generate nascent protein with distinct functional roles to mature RAB13 at specific subcellular sites, thus achieving tight spatial compartmentalisation of RAB13-mediated membrane remodelling. As predicted, such coupling of polarised *RAB13* mRNA localisation with local translation was confirmed in EC protrusions upon detection of nascent protein using puromycinilation-proximity ligation assays (Puro-PLA) (tom Dieck *et al*, 2015). ECs were cultured on Transwells and cell bodies removed prior to pulse labelling with puromycin to exclude detection of nascent proteins transported from the cell body to protrusions (Fig 5A). Isolated EC protrusions readily incorporated puromycin, which could be blocked upon pre-incubation with the translation inhibitor anisomycin (Fig EV4), indicating active protein translation at the leading edge of migrating ECs. Importantly, Puro-PLA on isolated EC protrusions using antibodies recognising puromycin and RAB13 revealed numerous distinct punctae corresponding to newly synthesised RAB13, unlike anisomycin pre-treated and antibody-free controls (Fig 5B and C). Hence, polarised targeting of *RAB13* mRNA to motile cell protrusions drives local RAB13 translation. Moreover, spatial control of mRNA polarisation and local translation was coupled to regional compartmentalisation of RAB13 protein function, as loss of endogenous *RAB13* specifically disrupted filopodia dynamics only at distal sites of mRNA targeting (Fig 5E–H). The siRNA-mediated knockdown of RAB13 expression (Fig 5E) did not perturb *RAB13*-independent filopodia at proximal regions in ECs (Fig 5H), but significantly depleted filopodia numbers at distal sites, as is particularly obvious in Fig 5F. Consequently, ECs exhibited overall reduced numbers of filopodia upon *RAB13* knockdown (Fig 5G). This was not simply a consequence of spatial targeting of protein to the leading edge, as immunofluorescence assays revealed that RAB13 was homogeneously distributed throughout migrating cells (Fig 5D), indicating that in contrast to the mRNA encoding it, RAB13 steady-state protein is not polarised. Alternatively, it was the location of *RAB13* mRNA itself that defined the domain of RAB13-dependent filopodia dynamics, as excision of the LE and

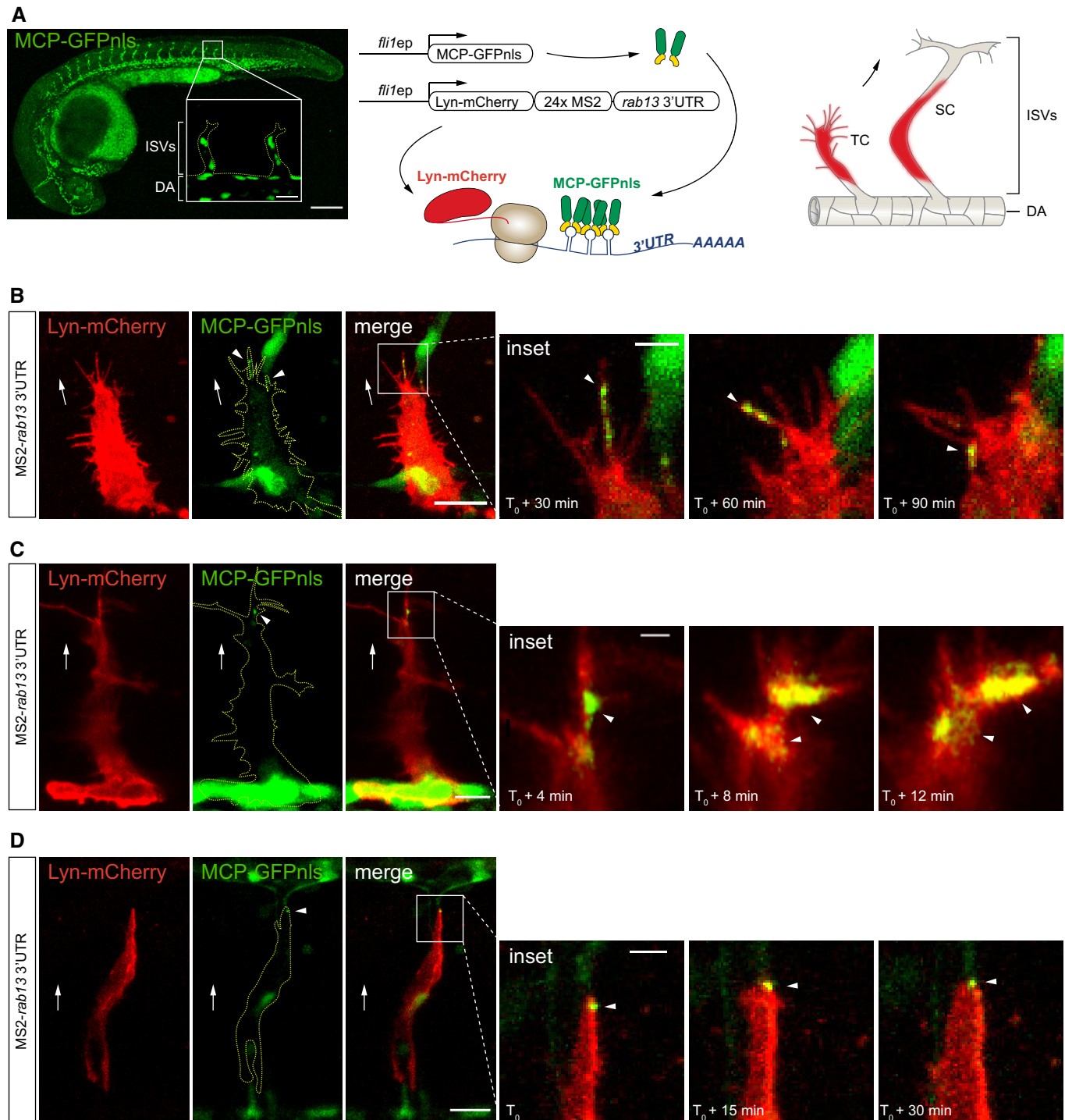

**Figure 6. The 3′UTR of *rab13* targets mRNA to endothelial cell protrusions *in vivo*.**

A Left: *Tg(fli1ep:MCP-GFPnls)* zebrafish embryo at 26 h post-fertilisation (hpf) displaying vascular-specific expression of MCP-GFPnls. Inset shows the nuclear expression of MCP-GFPnls in the intersomitic vessels (ISVs) sprouting from the dorsal aorta (DA). Middle: scheme depicts the *in vivo* MS2 system strategy with *fli1* enhancer/promoter (*fli1ep*)-driven expression of reporter constructs, simultaneous translation of Lyn-mCherry reporter and binding of MCP-GFPnls to 24xMS2-*rab13* 3′UTR. Right: scheme illustrates ISV cells expressing Lyn-mCherry imaged in panels B–D. TC: tip cell; SC: stalk cell.

B–D Time-lapse microscopy of *Tg(fli1ep:MCP-GFPnls)* tip and stalk cells displaying mosaic expression of Lyn-mCherry-24xMS2-*rab13* 3′UTR in ISV cells.

Data information: $T_0$ = 24 hpf (C), 28 hpf (B), 48 hpf (D). Arrowheads indicate non-nuclear localisation of MCP-GFPnls; arrows indicate direction of ISV sprouting; yellow dashed lines outline ISV (A) or ISV cell (B-D) borders; scale bars = 200 μm (A), 20 μm (B, D) and 10 μm (C); scale bars in insets = 20 μm (A), 5 μm (B, D) and 2 μm (C).

diffuse mislocalisation of *RAB13* mRNA was sufficient to drive ectopic depolarised filopodia (Fig 4D–G). Hence, the site of *RAB13* mRNA localisation, translation and protein function appears to be tightly spatially coupled in migrating cells. Consequently,

polarisation of *RAB13* transcript forms a molecular compass that achieves precise subcellular compartmentalisation of protein function, defines a polarised domain of filopodia extension and ultimately orients motile cell polarity (Fig 5I).

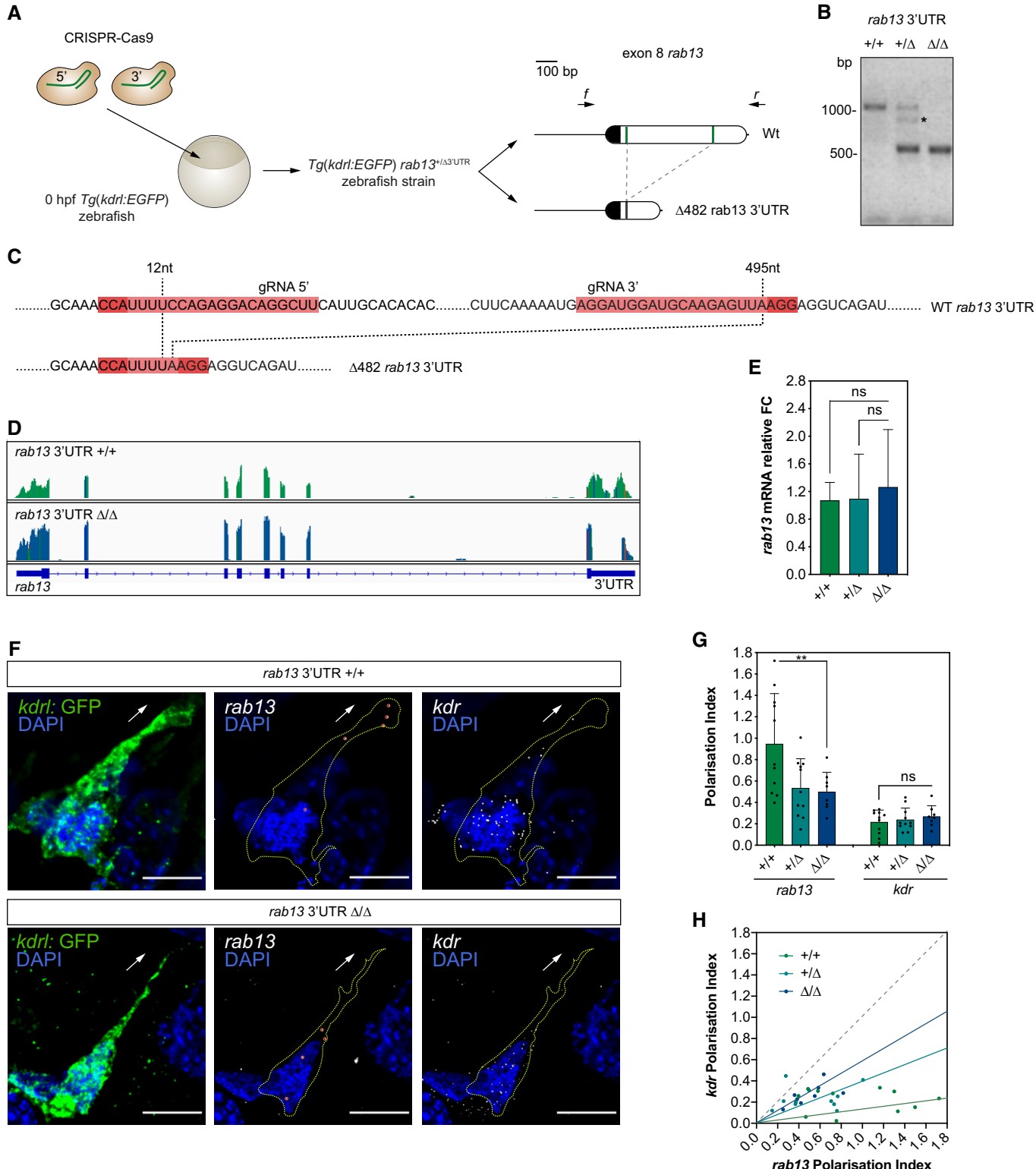

**Figure 7.**

◀

**Figure 7.  CRISPR-Cas9 editing of the zebrafish *rab13* 3′UTR perturbs mRNA polarisation.**

A  CRISPR-Cas9 strategy to generate the *Tg(kdrl:EGFP) rab13*[+/Δ3′UTR] zebrafish strain. The wild-type (Wt) *rab13* exon 8 is represented with its coding sequence in dark and the 3′UTR in clear boxes; the 5′ and 3′ gRNA-targeted regions are represented with green lines. Arrows: relative positions of the forward (*f*) and reverse (*r*) PCR primers used to identify animals with CRISPR-Cas9-mediated deletions (Δ) in the *rab13* 3′UTR.

B  Representative genotyping PCR demonstrates the band size shift in zebrafish harbouring a Δ482 *rab13* 3′UTR. Asterisk marks a heteroduplex formed between Wt and Δ482 *rab13* 3′UTR PCR amplicons.

C  Detailed DNA sequence depicting nucleotide positions within the Wt and Δ482 *rab13* 3′UTR.

D  RNAseq mapped reads depicting *rab13* exon usage in *Tg(kdrl:EGFP) rab13*[+/+] and *rab13*[Δ3′UTR/Δ3′UTR] zebrafish embryos. Coloured lines indicate SNPs.

E  qPCR analysis of *rab13* mRNA levels in individual 26–28 hpf clutch-matched sibling embryos ($n \geq 9$ embryos; ns: not significant; Kruskal–Wallis test with Dunn's correction).

F  smFISH detection of *rab13* and *kdr* mRNA in cultured GFP-expressing endothelial cells extracted from 48 hpf *Tg(kdrl:EGFP) rab13*[+/+] and *rab13*[Δ3′UTR/Δ3′UTR] zebrafish embryos.

G  Polarisation Index (PI) of *rab13* and *kdr* detected by smFISH in individual zebrafish cells ($n \geq 8$ cells; **$P < 0.01$, ns: not significant; one-way ANOVA with Bonferroni's correction).

H  *rab13* PI plotted against respective *kdr* PI. The slope of the coloured lines represents the average *rab13/kdr* PI ratio; the dashed grey line represents a 1:1 ratio ($n \geq 8$ cells).

Data information: +/+, +/Δ and Δ/Δ represent *Tg(kdrl:EGFP) rab13*[+/+], *rab13*[+/Δ3′UTR] and *rab13*[Δ3′UTR/Δ3′UTR] embryos, respectively (E, G, H). Arrows indicate orientation of RNA localisation; yellow dashed lines outline cell borders; red circles highlight smFISH spots; scale bars = 10 μm (F). Bar charts are presented as means ± s.d.

## The 3′UTR of *rab13* targets mRNA to endothelial cell protrusions *in vivo*

Although a widespread phenomenon, the functional role for localised mRNAs in tissue migration and vertebrate morphogenesis remains unexplored. Hence, having defined a key role for mRNA polarisation in the spatial control of EC behaviour *in vitro*, we then sought to define the broader relevance of this phenomenon to modulation of tissue dynamics *in vivo*. The production of polarised filopodia protrusions is a characteristic hallmark of motile endothelial tip cells, which lead new blood vessel branches during angiogenesis (Gerhardt *et al*, 2003; Isogai *et al*, 2003). As such, using live-cell imaging approaches in the zebrafish model system, we probed the function of *rab13* mRNA polarisation in the control of tip cell behaviour and angiogenesis *in vivo*. Firstly, we generated a novel vascular-specific MCP-GFPnls transgenic strain, *Tg(fli1ep:MCP-GFPnls)*, and monitored the targeting dynamics of a MS2-tagged *rab13* 3′UTR reporter during intersegmental vessel (ISV) angiogenesis (Isogai *et al*, 2003; Fig 6A). Dynamic accumulation of MCP-GFPnls adjacent to or within filopodia at the leading edge of ISV tip cells revealed that *rab13* mRNA localisation *in vivo* closely mirrored the 3′UTR-driven polarisation of human *RAB13* mRNA *in vitro* (Fig 6B and C; Movies EV2 and EV3). Thus, the targeting function of the *RAB13/rab13* 3′UTR is highly conserved, despite rather low sequence conservation (Fig EV5B and C). Importantly, polarised localisation of MCP-GFPnls was not observed in the absence of the MS2-tagged *rab13* 3′UTR (Fig EV5A), confirming the presence of LEs within this region. Moreover, the polarised targeting of *rab13* mRNA was retained in less-motile ISV stalk cells, which trail tip cells, although at less dynamic and more discrete foci (Fig 6D; Movie EV4). Hence, the polarised targeting of *RAB13/rab13* mRNA in motile cells is highly conserved between species and in tissues.

## CRISPR-Cas9 editing of the zebrafish *rab13* 3′UTR perturbs mRNA polarisation

Next, we sought to determine whether *rab13* mRNA localisation *in vivo* is functionally implicated in blood vessel spouting. Similar to *in vitro* experiments, we performed CRISPR-Cas9-mediated

excision of a fragment within the *rab13* 3′UTR locus (Δ3′UTR) to further confirm the presence of LEs and their potential role in *rab13* mRNA polarisation *in vivo*. Microinjection of zebrafish embryos with CRISPR-Cas9 ribonucleoprotein complexes targeting exon 8 of *rab13* was sufficient to generate germline *rab13*[Δ3′UTR/Δ3′UTR] mutants lacking 482-nt of the *rab13* 3′UTR, as confirmed by sequencing (Figs 7A–C and EV2C). Additional RNAseq analysis of mutant embryos verified that overall splicing of *rab13* mRNA was unaffected by genomic excision of the LE (Fig 7D) and confirmed the high specificity of CRISPR-mediated excision, as no nucleotide mismatches were observed at any putative low-frequency off-target sites (Fig EV2D). Moreover, genomic excision of the *rab13* LE had no effect on mRNA levels (Fig 7E), although protein levels could not be tested due to a lack of good antibodies against zebrafish Rab13. Importantly, smFISH applied to explanted endothelial cells from dissociated *rab13*[Δ3′UTR/Δ3′UTR] mutant embryos confirmed that *rab13* mRNAs lacking these LEs were more diffusely distributed (Fig 7F–H), similar to observations in human ECs (Fig 3H–J). In contrast, control *kdr* mRNAs displayed unperturbed PI measurements in *rab13* mutant versus Wt ECs (Fig 7F and G). Hence, we reveal a previously unappreciated and conserved role for 3′UTR LEs in the dynamic polarisation of *rab13* mRNA during cell migration.

## *rab13* mRNA polarisation orients blood vessel morphogenesis

During ISV branching, migrating tip cells must make key directional decisions, particularly when negotiating the multi-tissue junction of the horizontal myoseptum (Lu *et al*, 2004; Torres-Vazquez *et al*, 2004; Lamont *et al*, 2009) (HM; Fig EV5D). Hence, we hypothesised that mRNA localisation-mediated orientation of EC filopodia may indeed generate spatial cues that direct vascular tissue movement. Consistent with a key role for mRNA polarisation in the spatial coordination of vascular morphogenesis, live-cell imaging of ISVs branching in Wt and *rab13*[Δ3′UTR/Δ3′UTR] embryos revealed that loss of *rab13* polarisation severely perturbed tip cell path-finding decisions (Fig 8A and B). Unlike ISVs in Wt and *rab13*[+/Δ3′UTR] embryos that efficiently negotiated their way past the HM position, ISVs in *rab13*[Δ3′UTR/Δ3′UTR] mutants struggled with this directional decision,

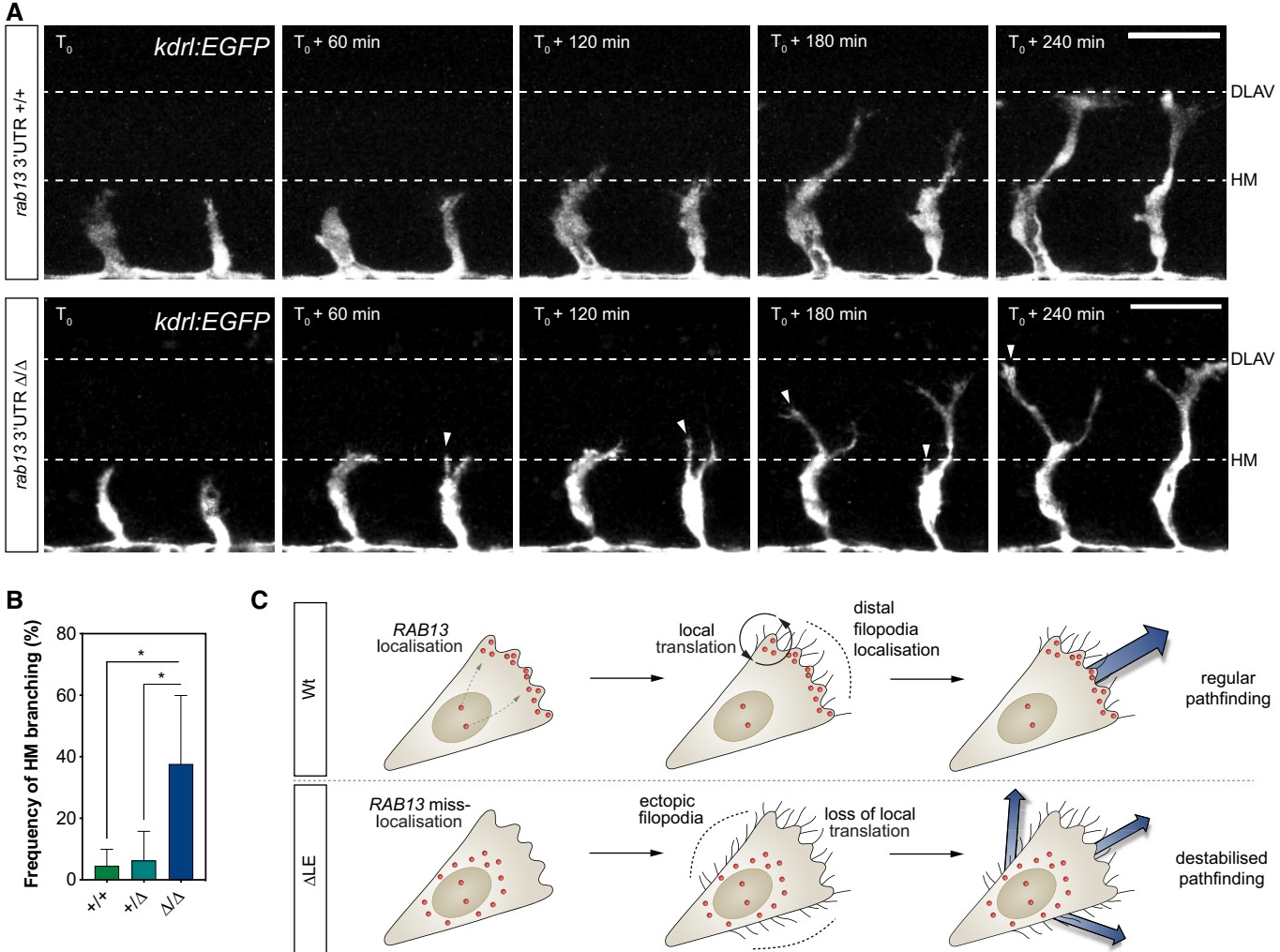

**Figure 8. *rab13* mRNA polarisation orients blood vessel morphogenesis.**

A   Time-lapse confocal microscopy of representative *Tg(kdrl:EGFP) rab13*^+/+^ and *rab13*^Δ3′UTR/Δ3′UTR^ embryos. DLAV: dorsal longitudinal anastomotic vessel; HM: horizontal myoseptum.

B   Frequency of ISV ectopic branching occurring at the HM (*n* = 4 experiments; \*P < 0.05; one-way ANOVA with Bonferroni's correction).

C   Illustration of the role for *RAB13* mRNA localisation, local translation and compartmentalisation of RAB13 function in defining the orientation of EC filopodia dynamics, motile EC polarity and blood vessel pathfinding.

Data information: $T_0$ = 25 hpf; arrowheads indicate extra branches emerging from the main ISVs at the HM position; scale bars = 50 μm (A). +/+, +/Δ and Δ/Δ represent *Tg(kdrl:EGFP) rab13*^+/+^, *rab13*^+/Δ3′UTR^ and *rab13*^Δ3′UTR/Δ3′UTR^ embryos, respectively (B). Bar chart is presented as means ± s.d.

resulting in a sevenfold increase in tip cells exhibiting ectopic misdirected branches (Fig 8A and B). Of interest, *rab13*^Δ3′UTR/Δ3′UTR^ mutants were viable with no detectable gross defects in other embryonic or vascular tissues, indicating a highly specific ISV phenotype. More importantly, *rab13* mRNA stability was unperturbed in *rab13*^Δ3′UTR/Δ3′UTR^ mutant embryos (Fig 7E), indicating that observed defects were not due to decreased *rab13* expression but a consequence of perturbed mRNA localisation. Thus, we provide the first *in vivo* evidence that spatial targeting of mRNAs and precise compartmentalisation of protein function generate key directional cues that orient motile cells during vertebrate tissue morphogenesis (Fig 8C).

# Discussion

Whilst it is well established that numerous transcripts are targeted to the leading protrusions of motile cells (Herbert & Costa, 2019), how this regulates translated protein function, its importance for cell migration and the *in vivo* relevance of this phenomenon are poorly understood. Here, using gene editing to modulate subcellular mRNA targeting, we reveal that tight spatial coupling of mRNA localisation, translation and protein function achieves precise subcellular compartmentalisation of protein action and prevents ectopic protein functionality at inappropriate subcellular loci. We find that such mRNA-mediated spatial compartmentalisation of RAB13 activity

serves to define a polarised domain of filopodia remodelling that orients motile cells. Moreover, using unique tissue-specific reporters of mRNA localisation in intact vertebrates, alongside gene editing, we uncover a key role for rab13 mRNA localisation in the coordination of cell pathfinding during tissue morphogenesis. Hence, we define mRNA polarisation as a novel paradigm for the spatial control of motile cell polarity and oriented tissue movement in vivo.

Moreover, our findings lend weight to recent observations that newly synthesised protein can have a distinct functionality to pre-existing protein (Kim et al, 2020). Considering that steady-state RAB13 protein is homogeneously distributed in migrating ECs, our work indicates that nascent protein must have distinct functional properties to the mature protein pool that enables tight spatial coupling of RAB13 translation and local filopodia remodelling. Indeed, in a parallel study published back-to-back with our work, Moissoglu et al (2020) show that nascent RAB13 co-translationally interacts with its exchange factor, RABIF, to drive local activation of newly synthesised protein in MDA-MB-231 cells. In their report, Moissoglu et al (2020) demonstrate that modulation of mRNA localisation using antisense oligonucleotides did not disrupt RAB13 expression, translation or steady-state protein localisation, but shifted the site and levels of co-translational RAB13 activation. Consequently, spatial manipulation of RAB13 mRNA targeting fundamentally defined the location of RAB13 protein action and perturbed cell protrusion in migration. Hence, the work by Moissoglu et al (2020) provides a convincing mechanistic basis for how RAB13 localisation ultimately orients motile cell polarity and tissue movement in vivo, via precise spatial control of co-translational interactions with exchange factors that define the site of GTPase activation.

Considering that RAB13 is one of only five mRNAs exhibiting conserved polarisation in all cell types tested, the function of RAB13 mRNA targeting may be a universally conserved mechanism for spatial coordination of complex morphogenetic events. Moreover, it is striking that all five-cluster k 5 mRNAs encode highly dynamic membrane trafficking and/or small GTPase-regulating proteins, all known to modulate cell motility. Hence, for classes of proteins normally constant in motion, mRNA polarisation may be essential to spatially compartmentalise and precisely fix their site of function. It is tempting to speculate that cluster k 5 mRNA co-targeting and localised translation may participate in a coordinated effort to modulate actin dynamics and/or membrane protrusion at the leading edge. As such, perturbation of the localisation mechanisms that transport RAB13 and other cluster k 5 mRNAs may be expected to generate a more acute phenotype than perturbation of RAB13 targeting alone. Indeed, other studies have provided evidence that mRNAs sharing subcellular compartments can encode subunits of common protein complexes involved in actin remodelling (Mingle et al, 2005) or components of related chemotaxis pathways (Hotz & Nelson, 2017). Nevertheless, the functional role of mRNA co-localisation in the context of cell migration remains largely elusive. As ever-increasing technological advances continue to unveil the nature of compartmentalised transcriptomes, this will shed further light on the patterns of co-localised mRNAs (Eng et al, 2019; Xia et al, 2019) and protein synthesis (Chouaib et al, 2020) that likely underpin key events directing cell migration.

Whilst it is well established that polarised trafficking of RAB13 and other cluster k 5 mRNAs are attributed to their interaction with APC, a microtubule plus-end associated protein that escorts mRNAs

to the leading edge of motile cells (Mili et al, 2008; Wang et al, 2017), the function of APC itself in cell migration is unclear. Despite reports on the importance of APC for EC migration (Harris & Nelson, 2010), knockdown of APC does not impact protrusion formation during migration in all cell types (Mili et al, 2008). Yet, from our work and that of Moissoglu et al (2020), it is clear that the function of RAB13 mRNA polarisation in the coordination of cell movement is itself conserved between distinct cell types and organisms. As such, a full explanation for the context dependency of APC function has yet to be defined, but may indicate differences in the mode of motility employed by distinct cell types, the types of dynamic protrusions employed (be they RAB13-dependent or not) or even hint at the use of currently unknown APC-independent mRNA transport mechanisms in cell migration.

Finally, this work reveals an unexpected spatial diversity to identified clusters of polarised mRNAs. Considering our observations that the sites of mRNA targeting and protein function are tightly coupled, this raises the exciting possibility that other distinct mRNA distributions, such as the perinuclear localisation of cluster k 7, reflect even broader functionalities for compartmentalised gene expression/function in the coordination of diverse aspects of tissue development, health and disease.

# Materials and Methods

### Zebrafish husbandry

Zebrafish were grown and maintained according to UK Home Office regulation guidelines, and all studies were approved by the University of Manchester Ethical Review Board.

### Embryo micro-injections and generation of zebrafish strains

To generate the transgenic zebrafish strain Tg(fli1ep:MCP-GFPnls) using Tol2 transposon transgenesis, 32 pg of Cerulean-H2B: basfli1ep:MCP-GFPnls Tol2-based plasmid was co-injected with 32 pg Tol2 mRNA into one-cell stage AB zebrafish embryos. The next day, embryos with mosaic GFP expression were selected, raised to adulthood and then outbred to AB zebrafish to identify founders with germline transmission of the transgene. Adult Tg(fli1ep:MCP-GFPnls) were inbred, and one-cell stage embryos were co-injected with 32 pg of Cerulean-H2B:basfli1ep:Lyn-mCherry-24xMS2-rab13-3′UTR Tol2-based plasmid and 32 pg Tol2 mRNA for mosaic expression analysis.

The mutant rab13 Δ3′UTR strain was generated with CRISPR-Cas9 tools. One-cell stage Tg(kdrl:EGFP)$^{s843}$ embryos (Jin et al, 2005) were injected with 150 pg of each in vitro transcribed gRNA and co-injected with 150 pg Cas9 NLS nuclease (New England Biolabs). Embryos were raised to adulthood and outbred to AB zebrafish to identify founders with germline transmission deletions in the rab13 3′UTR. Heterozygous animals harbouring a 482-nucleotide deletion in the rab13 3′UTR (Tg(kdrl:EGFP)$^{s843}$ rab13$^{+/Δ3′UTR}$) were in-crossed, and the resulting embryos were used for live-cell imaging analysis.

### gRNA generation and in vitro transcription

The online CRISPRscan tool (Moreno-Mateos et al, 2015) was used to design gRNAs targeting the zebrafish 3′UTR region in the

*rab13* locus (Table EV4) and to determine off-target loci. Next, 0.3 μM oligonucleotides comprising the target sequences (flanked by the T7 promoter and the Tail annealing sequence) were mixed with 0.45 μM Tail primer (Table EV4) and PCR-amplified with Platinum Pfx DNA Polymerase (Thermo Fisher Scientific) in a T100 thermal cycler (Bio-Rad). The following cycling conditions were used: 1 cycle of initial denaturation at 94°C for 10 min, 30 cycles of denaturation at 94°C for 30 s, annealing at 45°C for 30 s, extension at 68°C for 30 s and a final extension cycle at 68°C for 7 min. Subsequently, 200 ng of PCR-amplified templates was used to transcribe gRNAs using a MEGAshortscript T7 Transcription Kit (Thermo Fisher Scientific), following the manufacturer's recommendations.

To synthesise Tol2 mRNA, 1 μg NotI-linearised pCS2-TP plasmid was transcribed using a SP6 mMESSAGE mMACHINE kit (Thermo Fisher Scientific) according to the manufacturer's protocol.

### Embryo genotyping

Genomic DNA was extracted by incubating either whole embryos or embryo heads in lysis buffer (10 mM Tris–HCl pH 8, 1 mM EDTA, 80 mM KCl, 0.3% NP40, 0.3% Tween) containing 0.5 μg/μl Proteinase K (Promega) at 55°C for 1–2 h, followed by a denaturation step at 95°C for 15 min in a T100 thermal cycler. Genotyping PCR was performed using 2 μl genomic DNA, 0.4 μM zebrafish genotyping primers (Fig 7A and Table EV4) and 1× MyTaq Red DNA Polymerase (Bioline) according to the manufacturer's protocol in a T100 thermal cycler. PCRs were resolved in 1% agarose (Bioline) gels containing 0.5 μg/ml ethidium bromide (Sigma) for analysis. PCR products were cloned into TOPO-TA vectors (Thermo Fisher Scientific) according to the manufacturer's protocol and analysed via Sanger sequencing on an ABI 3730 device.

### Cell culture, scratch wound and co-culture angiogenesis assays

Trunks of 26–48 hpf embryos were incubated in trypsin–EDTA solution (Sigma) at 28°C for 15 min. Trypsinisation was quenched with complete L-15 medium (Sigma) containing 10% foetal bovine serum (FBS, Sigma) and 10 U/ml–100 μg/ml penicillin–streptomycin (Sigma). Cells were pelleted at 376 *g* for 5 min at room temperature (RT), resuspended in complete ECGM2 (PromoCell) and cultured on fibroblast-coated coverslips. Zebrafish cells were cultured in 24-well plates and maintained at 28°C for 18 h.

HUVECs (PromoCell) were cultured in complete ECGM2 (PromoCell) in gelatin-coated (Millipore) dishes. Human pulmonary fibroblasts (HPF; PromoCell) were cultured in M199 (Thermo Fisher Scientific) containing 10% FBS, 50 μg/ml gentamycin (Sigma) and 50 ng/ml amphotericin (Sigma). Brain endothelial cells (bEnd.3 and bEnd.5) were cultured in DMEM (Sigma) supplemented with 10% FBS, 10 ng/ml recombinant human VEGF-A (PeproTech) and 10 U/ml–100 μg/ml penicillin–streptomycin.

For scratch wound assays, HUVECs cultured on gelatin-coated coverslips were grown to confluence and used in scratch wound assays as described elsewhere (Liang *et al*, 2007).

Co-cultures of HUVEC and HPF and the corresponding siRNA-mediated knockdown experiments were performed as previously described by Hetheridge *et al* (2011).

### CRISPR-Cas9 cell editing and cell transfections

The online Alt-R CRISPR-Cas9 design tool (https://eu.idtdna.com) was used to design crRNAs targeting the RAB13 3′UTR locus and to determine off-target loci. HUVECs were transfected with Alt-R CRISPR-Cas9 ribonucleoprotein complexes (Integrated DNA Technologies) targeting the 90–282-nt localisation element within the 3′UTR. Briefly, each sequence-specific crRNA (Table EV4) was mixed with tracrRNA at 1:1 50 μM, incubated at 95°C for 5 min in a T100 thermal cycler and allowed to cool to RT for 60 min. Next, 12 μM each crRNA:tracrRNA (gRNA) was incubated with 20 μM Alt-R Cas9 nuclease in PBS (Sigma) at RT for 20 min to form ribonucleoprotein complexes and mixed with $500 \times 10^3$ HUVECs. Additionally, 2 μg pmaxGFP Vector (Lonza) was included in the HUVEC-ribonucleoprotein mix to identify transfected cells. Transfections were performed in a Nucleofector 2b Device (Lonza), using a HUVEC Nucleofector Kit (Lonza) according to the manufacturer's instructions, and the cells were further cultured for 72 h. Afterwards, single GFP-expressing cells were isolated in a FACSAria Fusion cell sorter (BD Biosciences) into gelatin-coated 96-well plates to grow individual clones. Genomic DNA was extracted from expanded HUVEC clones and PCR-analysed with sequence-specific primers (Fig 3A and Table EV4) as described for zebrafish embryo genotyping. Clones with either biallelic deletion of the localisation element (ΔLE) or with the full-length *RAB13* 3′UTR (Wt) were maintained until passage 6, and three clones from each genotype were used for analysis.

Knockdown experiments were performed with ON-TARGETplus Non-targeting Control or *RAB13* siRNAs (Horizon) using GeneFECTOR (VennNova) as previously described (Hetheridge *et al*, 2011).

For *in vitro* MS2 experiments, bEnd.3 or bEnd.5 cells were transfected with pcDNA3-Lyn-mCherry, pCS2-MCP-GFPnls and different versions of pcDNA3-*HBB*-24XMS2SL-*RAB13* 3′UTR. Briefly, $100 \times 10^3$ cells/well cultured in 6-well plates were transfected with 0.5–1 μg each plasmid DNA using Lipofectamine 2000 or Lipofectamine 3000 following the manufacturer's protocol (Thermo Fisher Scientific) and analysed 48 h later.

### Transwell assays and cell body/protrusion fractionation

Transwell experiments to segregate cell bodies and protrusions were performed as described elsewhere (Mili *et al*, 2008), with the following modifications: $1.5 \times 10^6$ HUVECs were cultured for 2 h in 24-mm Transwells (Costar), containing 3-μm-pore polycarbonate membranes, in M199 (Thermo Fisher Scientific) supplemented with 1% FBS. Subsequently, 25 ng/ml VEGF-A was added to the lower chambers to promote cell migration over the next hour. Whilst only 1 Transwell was used for the cell body fraction, 2 Transwells were used to harvest each HUVEC protrusion sample.

### RNA isolation, qPCR and RNAseq

Embryo and cell-derived RNA was isolated using a RNAqueous-Micro Kit (Thermo Fisher Scientific) according to the manufacturer's protocol. For gene expression analysis, cDNA was synthesised with a High-Capacity RNA-to-cDNA Kit (Thermo Fisher Scientific) following the manufacturer's protocol.

qPCR experiments were performed with 1–2 μl cDNA, 0.25 μM gene-specific primers (Table EV4) and 1× Power SYBR Green Master

Mix (Thermo Fisher Scientific) in a StepOne Real-Time PCR System (Applied Biosystems). *GAPDH* expression was used to normalise gene expression levels, and the relative mRNA levels were analysed with the $2^{-\Delta\Delta CT}$ method.

For RNAseq, quality and integrity of RNA samples obtained from HUVEC cell bodies and protrusions were assessed using a 2200 TapeStation (Agilent Technologies). Next, RNAseq libraries were generated using the TruSeq Stranded mRNA assay (Illumina) according to the manufacturer's protocol. Adapter indices were used to multiplex libraries, which were pooled prior to cluster generation using a cBot instrument. The loaded flow cell was then paired-end-sequenced (76 + 76 cycles, plus indices) on an Illumina HiSeq 4000 instrument, and the output data were demultiplexed (allowing one mismatch) and BCL-to-Fastq conversion performed using Illumina's bcl2fastq software, v2.17.1.14. Sequence adapters were removed, and reads were quality trimmed using Trimmomatic v0.36 (Bolger *et al*, 2014) (Transwell samples) or BBDuk (part of the BBMap suite; v36.32) (HUVEC and zebrafish CRISPR-Cas9 experiments). Processed reads from the human-derived samples were mapped against the reference human genome (hg38) using STAR v2.5.3/2.7.2b (Dobin *et al*, 2013), and counts per gene were calculated using annotation from GENCODE v30/32 (http://www.gencodegenes.org/). Zebrafish-derived samples were mapped against the reference assembly GRCz11 and gene annotation from Ensembl v99. Normalisation and differential expression was calculated with Bioconductor package DESeq2 v1.24 (Transwell samples), and RNAseq mapped reads were visualised with Jalview v2.11.0 (Waterhouse *et al*, 2009) (HUVEC and zebrafish CRISPR-Cas9 experiments).

### smFISH

Zebrafish cells and HUVECs were fixed in methanol-free 4% formaldehyde (Thermo Fisher Scientific) and used in smFISH assays. Briefly, cells were permeabilised with 70% ethanol at RT for 1 h or 4°C overnight, washed with smFISH wash buffer (2× SSC, 10% formamide) and incubated with smFISH probes (Table EV5) in smFISH hybridisation buffer (10% dextran sulphate, 2× SSC, 10% formamide) at 37°C overnight. Afterwards, cells were washed with smFISH wash buffer twice at 37°C for 30 min, washed once with 2× SSC for 10 min, counterstaining with 1 µg/ml DAPI (Sigma) and washed twice with PBS for 5 min at RT. Coverslips were air-dried and mounted on microscope slides with ProLong Gold Antifade Mountant (Thermo Fisher Scientific). All probes targeting protrusion-enriched mRNAs were designed with Stellaris Probe Designer (LGC Biosearch Technologies), synthesised and labelled with Quasar 570 or Quasar 670 (LGC Biosearch Technologies). Alternatively, probes were synthesised with an upstream FLAP sequence (CCTCCTAAGTTTCGAGCTGGACTCAGTG) (Tsanov *et al*, 2016) and annealed to a complementary FLAP probe labelled with Alexa 594 (Integrated DNA Technologies). Co-hybridisation experiments were carried out with predesigned *GAPDH* probes labelled with Quasar 670 (HUVECs) or *kdr* probes labelled with Quasar 570 (zebrafish cells) (LGC Biosearch Technologies).

### Puro-PLA and immunofluorescence (IF)

For Puro-PLA, cell bodies of HUVECs cultured in Transwells were scraped off and remaining protrusions were exposed to 3 µM

puromycin (Sigma) added to lower chambers for 6 min. In translation inhibition experiments, 40 µM anisomycin (Sigma) was added to the lower Transwell chamber 30 min before cell body removal and 6 min after cell body removal together with 3 µM puromycin. Subsequently, HUVEC protrusions grown in Transwell membranes were fixed in methanol-free 4% formaldehyde, removed from the Transwell inserts and used in Puro-PLA experiments as described elsewhere (tom Dieck *et al*, 2015). Following the Puro-PLA protocol, Transwell membranes were incubated for 20 min with 1:40 Alexa Fluor 488 Phalloidin (Thermo Fisher Scientific) in PBS, washed in Duolink wash buffer B (Sigma) and mounted on microscope slides with Duolink *In Situ* Mounting Medium containing DAPI (Sigma).

For IF experiments, cells and Transwell membranes containing protrusions were permeabilised in PBS containing 0.2–0.5% Triton X-100 (Sigma), blocked in 4% goat serum (Sigma) for 15 min and incubated with primary antibodies in blocking solution at 4°C overnight. Next, cells were washed in PBS containing 0.2% Tween, incubated with secondary antibodies at RT for 1 h, counterstaining with 1 µg/ml DAPI and washed again. Transwell membranes were further incubated with 1:40 Phalloidin Alexa Fluor 488 (Thermo Fisher Scientific) in PBS at RT for 20 min before washing. Cells and Transwell membranes were mounted with ProLong Gold Antifade Mountant (Thermo Fisher Scientific).

### Western blotting

Proteins were extracted with RIPA buffer (25 mM Tris–HCl pH 7.6, 150 mM NaCl, 1% NP-40, 1% sodium deoxycholate and 0.1% SDS) and quantified with Pierce BCA Protein Assay Kit (Thermo Fisher Scientific) following the supplier's recommendations. Samples were denatured with Laemmli buffer (250 mM Tris–HCl pH 6.8, 2% SDS, 10% glycerol, 0.0025% bromophenol blue, 2.5% β-mercaptoethanol) at 95°C for 5 min, loaded on 10% Mini-PROTEAN TGX precast protein gels (Bio-Rad) and separated in a Mini-PROTEAN Electrophoresis System (Bio-Rad). Proteins were transferred onto nitrocellulose membranes using a Trans-Blot Turbo Transfer System RTA Kit following the manufacturer's protocols (Bio-Rad). Subsequently, membranes were blocked in 5% milk (Sigma) or 5% BSA (Sigma) in TBS containing 0.1% Tween at RT for 1 h and incubated with primary antibodies at 4°C overnight. The next day, membranes were washed with TBS containing 0.1% Tween, incubated with secondary antibodies at RT for 1 h and washed again. Signal detection was carried out with SuperSignal West Dura Extended Duration Substrate (Thermo Fisher Scientific) according to the supplier's recommendations.

### Antibodies

Primary and secondary antibodies were used at the following concentrations: 1:1,600 mouse PECAM-1 89C2 (Cell Signaling Technology), 1:100 rabbit RAB13 (Puro-PLA, Millipore), 1:1,000 rabbit RAB13 (Western blotting and IF, Cambridge Bioscience), 1:3,500 mouse puromycin (Kerafast), 1:1,000 rabbit β-tubulin 9F3 (Cell Signaling Technology), 1:200 mouse ZO-1 1A12 (Thermo Fisher Scientific), 1:500 goat anti-mouse Alexa Fluor 488 or Alexa Fluor 568 (Thermo Fisher Scientific), 1:500 goat anti-rabbit Alexa Fluor 568 (Thermo Fisher Scientific), 1:5,000 goat anti-mouse HRP-linked

(Cell Signaling Technology) and 1:5,000 goat anti-rabbit HRP-linked antibody (Cell Signaling Technology).

## Plasmid construction

The pCS2-MCP-GFPnls plasmid used in *in vitro* MS2 system assays was generated excising a MCP-GFPnls fragment with SpeI and KpnI from pMS2-GFP, a gift from Robert Singer (Addgene plasmid # 27121) (Fusco *et al*, 2003), and subcloning it into a pCS2 + vector using the XbaI and KpnI sites.

To construct the Cerulean-H2B:*basfli1ep*:MCP-GFPnls Tol2-based plasmid for *in vivo* studies, MCP-GFPnls was amplified from pMS2-GFP with sequence 0.3 µM specific primers (Table EV4) and Platinum Pfx DNA Polymerase in a T100 thermal cycler. Subsequently, the PCR product was cloned into a pDONR221 P3-P2 using Gateway Technology (Thermo Fisher Scientific) according to the manufacturer's manual. The final Tol2-based construct was assembled into the pTol2Dest(R1R2) (Addgene plasmid # 73484) (Villefranc *et al*, 2007) using Gateway 3-fragment recombination with pE(L1L4)Cerulean-H2B in the first position, pE(R4R3)*basfli1ep* (De Bock *et al*, 2013) in the second position and pE(L3L2)MCP-GFPnls in the third position.

For *in vitro* MS2 system experiments, 3′UTRs were PCR-amplified from human genomic DNA with 0.3 µM sequence-specific primers (Table EV4) using Platinum Pfx DNA Polymerase or MyTaq Red DNA Polymerase (Bioline) in a T100 thermal cycler and the resulting PCR product was cloned using either Zero Blunt PCR or TOPO TA Cloning Kits (Thermo Fisher Scientific), following the manufacturer's manual. Next, the human *HBB* gene was PCR-amplified using 0.3 µM sequence-specific primers (Table EV4) and Platinum Pfx DNA Polymerase in a T100 thermal cycler and cloned into the NotI and BamHI sites of the pCR4-24XMS2SL-stable plasmid, a gift from Robert Singer (Addgene plasmid # 31865) (Bertrand *et al*, 1998). Subsequently, a multiple cloning site (MCS; Table EV4) was introduced into the BglII and SpeI sites of pCR4-*HBB*-24XMS2SL and the recombinant *HBB*-24XMS2SL-MCS sequence was subcloned into the pcDNA3 mammalian expression vector (Thermo Fisher Scientific) using the NotI and XbaI sites. Full length 3′UTRs were then subcloned into pcDNA3-*HBB*-24XMS2SL-MCS using NheI and XhoI/ApaI sites. Alternatively, truncated and deletion versions of the 3′UTRs were generated by PCR using 0.3 µM sequence-specific primers (Table EV4) and Platinum Pfx DNA Polymerase, Phusion High-Fidelity polymerase (NEB) or using QuikChange II Site-Directed Mutagenesis Kit (Agilent Technologies) following the manufacturer's instructions and introduced into the pcDNA3-*HBB*-24XMS2SL-MCS using the NheI and XhoI sites.

In order to generate the zebrafish MS2 system reporter construct, the 24XMS2SL cassette was firstly subcloned from pCR4-24XMS2SL-stable into a *kdrl*:Lyn-mCherry Tol2-based plasmid (Costa *et al*, 2016) using a BamHI site. Next, the zebrafish *rab13* 3′UTR was PCR-amplified with 0.4 µM sequence-specific primers (Table EV4) and MyTaq Red DNA Polymerase from zebrafish genomic DNA in a T100 thermal cycler and then subcloned into the Tol2 *kdrl*:Lyn-mCherry-24XMS2SL plasmid using NheI and BglII sites. The resulting Lyn-mCherry-24XMS2SL-*rab13* 3′UTR recombinant sequence was amplified with 0.3 µM sequence-specific primers (Table EV4) and Platinum Pfx DNA Polymerase in a T100 thermal cycler and

subcloned into a pDONR221 P3-P2 using Gateway Technology. Lastly, the final Tol2-based construct was assembled into the pTol2Dest(R1R2) using Gateway 3-fragment recombination with pE (L1L4)Cerulean-H2B in the first position, pE(R4R3)*basfli1ep* in the second position and Lyn-mCherry-24XMS2SL-*rab13* 3′UTR in the third position.

All plasmid maps and details are available upon request.

## Microscopy

Confocal time-lapse imaging of zebrafish embryos was carried out as previously described (Costa *et al*, 2016). MS2 system-transfected cells were live-imaged every 5 s in a Nikon A1R-inverted confocal microscope equipped with an Okolab incubation chamber, using a 60× objective. Fixed images of cultured cells and Transwell membranes were acquired on an Olympus IX83-inverted microscope using Lumencor LED excitation, either a 60×/1.42 PlanApo or a 100×/1.35 UplanApo objective and a Sedat QUAD (DAPI/FITC/TRITC/Cy5) filter set (Chroma 89000). The images were collected using a R6 (Qimaging) CCD camera with a Z optical spacing of 0.2 µm. Raw images were then deconvolved using the Huygens Pro software (SVI), and maximum intensity projections of these images were used for analysis.

## smFISH spot quantification, Polarisation Index and filopodia analysis

Processed smFISH images were used to calculate mRNA polarisation with the PI metric developed by Park *et al* (2012) and to assess mRNA spot number with FISH-quant (Mueller *et al*, 2013).

For the studies of filopodia distance to GFP signal in MS2 system movies, filopodia parameters (position, duration and frequency) of MS2 system-transfected cells were determined using Filopodyan plugin for FIJI (Urbancic *et al*, 2017). Only filopodia that emerged and retracted through the duration of the movies were analysed. Subsequently, the coordinates of GFP particles were extracted with the TrackMate plugin for FIJI (Tinevez *et al*, 2017). In order to generate control coordinates in each movie, the Lyn-mCherry channel was thresholded to generate regions of interest (ROI) and the FIJI built-in macro function "random" was used within the ROI at each frame. The Euclidean distances between the base of newly formed filopodia and both the nearest GFP particle and the control randomised coordinate were calculated.

## RNA motif enrichment analysis and Gene Ontology

Discovery of recurring motifs across the $k$ 5 mRNA 3′UTRs was carried out using MEME (Bailey & Elkan, 1994) (settings set to: mode - anr; nmotifs: 5; minw: 6; maxw: 50; objfun classic: markov_order 0). The only motif present in all 3′UTR was selected for downstream studies. In order to quantify the frequency mRNA within the remaining $k$-means clusters containing at least two repeats of the studied RNA motif, 3′UTR sequences were scanned in FIMO (Grant *et al*, 2011) using the position-specific probability matrix obtained in MEME (settings set to: match *P*-value < 1E-5).

Gene Ontology studies were performed using DAVID using the HUVEC-enriched mRNAS as background (Huang da *et al*, 2009a,b).

### Statistics and *k*-means clustering

All data are represented as means ± standard deviation. Statistical analysis of the data was carried out using GraphPad Prism software and RStudio. D'Agostino–Pearson or Shapiro–Wilk normality tests were applied to smFISH, Puro-PLA, filopodia, ISV branching data and RNA, protein levels to determine the appropriate statistical test. Statistical significance is reported for $P < 0.05$.

*k*-means clustering was performed in RStudio. Briefly, mRNA fold changes (FC) between cell bodies and protrusions of the cell types mentioned in the main text were obtained from the respective publications—NIH/3T3 fibroblasts (Wang *et al*, 2017), MDA-MB231 metastatic breast cancer cells (Mardakheh *et al*, 2015) and induced neuronal cells (Zappulo *et al*, 2017). mRNAs were included in the clustering analysis if they were enriched in HUVEC protrusions (FC > 1.6, FDR < 0.05) and expressed in all three cell types. The $\log_2$ FC values between cell fractions were extracted, scaled and centred, followed by *k*-means clustering ($k = 8$). The number of clusters was defined by the number of cell types and by the possible transcript statuses (enriched or depleted in each of the three cell types) $- 2^3 = 8$. The output is represented in heat maps using the $\log_2$ FC data prior to scaling and centring.

## Data availability

The RNAseq datasets presented in this study have been deposited to the Gene Expression Omnibus repository (https://www.ncbi.nlm.nih.gov/geo/) with the accession numbers GSE133055 and GSE155449.

RStudio scripts are available upon request.

**Expanded View** for this article is available online.

### Acknowledgements

We are grateful to E. Schuman and S. tom Dieck at the MPI for Brain Research, Frankfurt, Germany, for help and guidance setting up Puro-PLA assays. Moreover, we thank Rob Wilkinson at the University of Nottingham, UK, for access to his preliminary results for CRISPRi-mediated *rab13* knockdown in zebrafish. We also thank N. Papalopulu and group members at the University of Manchester, UK, for critical feedback, reagents and materials. We wish to thank members of the University of Manchester Biological Services, Genomic Technologies, Bioimaging Facilities and Flow Cytometry Facilities, for technical support. This work was funded by the Wellcome Trust (095718/Z/11/Z and 219500/Z/19/Z to S.P.H.), Wellcome Institutional Strategic Support Fund (204796/Z/16/Z to G.C.) and the British Heart Foundation (PG/16/2/31863 to S.P.H.).

### Author contributions

GC and SPH conceptualised the study. GC contributed to methodology. GC involved in formal analysis. GC, JB and NT investigated the study. GC and SPH wrote the original draft of the manuscript. GC and SPH wrote, reviewed and edited the manuscript. GC and SPH supervised the study. GC and SPH acquired the funding.

### Conflict of interest

The authors declare that they have no conflict of interest.

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
