## [Review Process File · The EMBO Journal]

RAB13 mRNA compartmentalisation spatially orients tissue morphogenesis

Shane Herbert, Guilherme Costa, Joshua Bradbury, and Nawseen Tarannum

DOI: 10.15252/embj.2020106003

Corresponding author(s): Shane Herbert (shane.herbert@manchester.ac.uk), Guilherme Costa (g.costa@qub.ac.uk)

Review Timeline:

Submission Date:	22nd Jun 20
Editorial Decision:	9th Jul 20
Revision Received:	25th Jul 20
Editorial Decision:	7th Aug 20
Revision Received:	8th Aug 20
Accepted:	14th Aug 20

Editor: Elisabetta Argenzio

Transaction Report:

The initial review process for this manuscript took place with another journal. The initial reviewers' comments and authors' responses for this article have been made available. All referees have been contacted and agreed to have their comments published with this article.

Referees' comments:

Referee #1 (Remarks to the Author):

The manuscript by Costa et al. reveals a functional role for subcellular localization of an endogenous mRNA (Rab13) in endothelial cells using elegant experiments (including gene editing of localization elements) in cell culture and in zebrafish. Although Rab13 mRNA was previously shown to localize in protrusions in cultured cells (by Mili and Nielsen and colleagues) the functional significance of this process was unclear. And while requirements for mRNA localization elements have previously been reported in other organisms, notably yeast and *Drosophila*, this study is the first such example in vertebrates. This is therefore an important study that is likely to be seen as a landmark in the mRNA localization field. The paper should also appeal more generally to those interested in cell migration and endothelial cell biology as the authors provide evidence that filopodia formation is guided by a local translation-based mechanism that prevents formation of these structures at ectopic sites (rather than an active mechanism to promote filopodia formation at one location). In addition to the advances summarized above, the authors also characterize other mRNA localization patterns in endothelial cells that pave the way for future work in this system.

I am enthusiastic about the manuscript but additional controls are required to back up some of the key conclusions. In addition, some clarification is needed to place the findings in the context of other studies on Rab13 mRNA localization.

Major points:

1. Rab13 appears to have different predicted splice isoforms (at least in human cells) and 3'UTR sequences can affect splice site selection (e.g. see <https://doi.org/10.1016/j.celrep.2019.05.083>). Defects in splicing could in principle account for the phenotypes observed, independently of mRNA localization, and it is therefore important that the authors check if patterns of splicing are altered by the deletion in the 3'UTR. The authors should also quantify Rab13 mRNA levels in wild-type and 3'UTR mutant HUVECs to strengthen their case that changes in overall mRNA levels do not contribute to the phenotypes in this system.
2. The consistent phenotypic outcomes of removing the localization elements in the Rab13 3'UTR in HUVECs and zebrafish (which requires different gRNA sequences) argues against off-target effects of the CRISPR/Cas9 procedure being responsible for the observed phenotypes. Nonetheless, more needs to be done to strengthen this argument, which is the central part of the study. Do the gRNAs used in each experiment have predicted off-target effects? If so, the authors should check that potential off-target sites are not edited in the cells/animals used. A powerful experiment, which I would class as desirable but not essential, would be to test if the defects caused by removal of the localization element make cells/animals particularly sensitive to partial knockdown of Rab13 (by incomplete siRNA or heterozygosity for a null mutation in the case of zebrafish). If this were the case, it would be very strong evidence for the specificity of the CRISPR/Cas9 phenotypes. Or did the authors recover fish with wild-type Rab13 alleles that went through the CRISPR/Cas9 procedure? A strong off-target effect might be expected to give phenotypes when this wild-type Rab13 allele is homozygous.
3. The importance of Rab13 mRNA localization for cell migration was previously challenged by the observation that APC is required for localization of this mRNA but not for cell migration (summarized in <http://www.jbc.org/cgi/doi/10.1074/jbc.R116.715193>). Costa et al. need to discuss this point. Is APC required for Rab13 mRNA localization in HUVEC cells? If so, might the requirements for APC-mediated Rab13 mRNA localization be specific for certain cell types? Is it known if APC is required for branching in the endothelial system in zebrafish?
4. Additional controls are needed for the PURO-PLA experiments. The authors should repeat the experiment when Rab13 is knocked down by siRNA in order to confirm the specificity of the antibody in these experiments (not just for western blotting). It would also be desirable to repeat this analysis in the cells that lack the region of the 3'UTR required for mRNA localization as this will allow a direct assessment of the link between mRNA localization and local translation.

Minor points:

1. How conserved is the region of the Rab13 3'UTR that mediates localization? An alignment of Rab13 sequences from different species (including human and zebrafish) should be provided in the Extended Data section.
2. The type of statistical tests used for each panel should be specified in the legends or a supplementary table (depending in the journal's policy). Currently, the methods state that different tests were performed depending on the data distribution, which is correct, but the reader needs to be able to understand which test was applied where.
3. The manuscript is mostly clearly written but in some sections it can be made much more concise. There are also several typos and spelling mistakes in the manuscript:

E.g.: miss-localization, longstading.
4. In the abstract, insert 'mRNA' between 'endogenous RAB13' and 'targeting' and revise 'endogenous gene-edited mRNAs', which is a potentially misleading term.
5. Page 5: "exclusive targeting of mRNA to sites of incipient filopodia formation in cellular protrusions". Is targeting really exclusive? Also, is the mRNA in cellular protrusions or in a broader region of the cell that is prone to form cellular protrusions? This is an important distinction when trying to understand the spatio-temporal aspects of the system.
6. Mili et al. recently reported on the relationship between Rab13 mRNA localization and translation (<https://elifesciences.org/articles/44752>). Although this work was in different cell lines to those used by Costa et al. the authors should consider citing relevant findings from this study.
7. Extended data Figure 2a: is the motif predicted to form a G-quadruplex?

Referee #3 (Remarks to the Author):

In this manuscript, the authors investigate how polarized localization of mRNAs can contribute to tissue morphogenesis. Through RNAseq analysis of endothelial cell bodies and protrusions, they identify transcripts that are selectively localized to cellular protrusions. Comparison with other cell types from published studies identified a core set of protrusion-localized genes, including RAB13, which they investigated further. They identify a conserved cis element in the 3' UTRs of these transcripts that suggested it is responsible for compartmentalization. Deletion of this element in RAB13 in human cells disrupts its localization and leads to defects in filopodial dynamics. Further investigation in zebrafish suggests this UTR element is conserved, required for cell compartmentalization, and is important for directional migration during endothelial sprouting.

Conceptually, this work is a potentially impactful move forward in functionally linking the significance of subcellular mRNA localization with a cellular and tissue morphogenetic role. However, there are significant experimental and technical gaps that leave the work incomplete. Most importantly, any data on how mRNA localization (and lack thereof) actually affects cell and tissue morphogenesis is completely lacking. Therefore, the studies as they currently stand are only suggestive and preliminary. Particular details are listed below.

Major concerns.

The computational work applied throughout the study is incompletely described and a bit weak. For example, RNAseq analysis of HUVEC protrusions only relies on duplicate samples and no statistical

values are given (e.g. in Fig 1b). This issue is further compounded by the k-means clustering, which shows a pretty clear cluster for the previously published data, but is much less convincing for HUVEC. It seems the authors could have easily identified this signature without the HUVEC analysis. If they insist on including the HUVEC data, appropriate statistical cut-offs should be applied, as well as any manipulations to apply normalization or batch effects to accurately compare the public data with their own. Similarly, it is unclear how the cis regulatory motif was identified. Was this by computational means? Is this significantly enriched in only the k5 protrusion-localized transcripts? What happens if they take a randomly selected set of transcripts and search for over-represented motifs? Presumably, their motif would be more likely to be found in other protrusion-localized transcripts, regardless of cell type - is this the case? What about conservation in other species?

Regarding the cis regulatory motif, the authors would greatly strengthen their study if they could more rigorously identify a minimal motif that was sufficient to cause localization (even if it needs to be in multiple copies). In this regard, finer deletion analysis/point mutations and heterologous assays would be welcome. This would allow for better assessment of its occurrence in other transcripts. Further, it may provide additional insights on mechanisms that contribute to its localization. In the absence of a consequential cellular or tissue morphogenesis phenotype for RAB13 (both of which are missing so far here), it may require disruption of the localization mechanism itself (i.e. whatever protein is binding to this motif) to perturb multiple effectors and see a phenotypic effect.

For the genome editing in HUVEC, I was always under the impression that isolating HUVEC clones was very challenging given that they are primary cells and will senesce over a limited number of passages. This raises some concerns about these experiments, especially since no controls were performed (e.g. parallel generation of clones using irrelevant sgRNA, or targeting downstream of the UTR, also a CDS mutation is important for comparison). Multiple clones would also be important. All of these particular controls are important to make sure the observed results are not arising due to some other issue with deriving HUVEC clones. It seems that this core set of polarization-localized genes is in many cell types. Perhaps it would be easier to pick the cell type (or better yet multiple types) that is most amenable to manipulation for these studies.

The HUVEC studies lack any assessment of migratory or protrusive activity. Are these affected by disrupting RAB13 localization? What about directionality? Persistence? Can they sprout in 3D matrix? Form tubes in matrigel? Etc. . . ? The authors need to assess if there is actually an effect on cell migratory behaviors, otherwise the relevance of their current findings is unclear.

The zebrafish studies suffer from some of the same issues. The authors need controls for the MS2 localization studies - e.g. a construct with a mutated or deleted LE motif and/or one with "irrelevant" UTR for comparison. It is also important to generate a coding sequence mutation for comparison to the UTR deletion. The authors should provide an alignment of the cis motif in fish and humans (other species would be good to see as well) to assess conservation. This may also aid in finding a minimal sufficient localization motif, as noted above.

Along similar lines, the phenotypic characterization of the zebrafish mutant is quite preliminary. Numerous questions and issues need to be addressed: Where and when is *rab13* expressed? Lots of different non-autonomous defects could lead to the observed phenotype. Therefore it is important to know what the overall phenotype of the embryo is. For the ISV defects, is there a defect in circulatory system patterning or function? How do defects compare to a full deletion of the coding sequence or a UTR deletion that does not remove the cis element? Regardless of cell type affected, experiments to address cell autonomy must be performed, eg. transplants or transgene-directed rescues. The latter approach on a complete *rab13* deletion allele would enable cell autonomous rescue with different UTR constructs and should be considered. I would note that if the goal of their efforts is to demonstrate that mRNA localization is important for tissue morphogenesis, then what the actual cell type is that exhibits a defect is inconsequential - but all of the aforementioned issues need to be addressed regardless.

Minor comments

Localization in the zebrafish. The smFISH experiments relied on explanted cells. Presumably sensitivity is an issue? Also, what is the localization of *rab13* protein in zebrafish? Obviously this is not

trivial to determine in zebrafish, but would be important confirmatory data. For MS2-visualized localization, what is the pattern in non-tip cells?

". . .monitored the dynamics of rab13 3'UTR mRNA. . ." is written as if you are monitoring endogenous transcript, but this is really the 3'UTR from a transgene. Should probably modify this to be more explicit.

Referee #4 (Remarks to the Author):

Summary:

Using an RNAseq screen the authors identified mRNAs enriched in protrusions of migrating HUVECs. Of 320 identified transcripts, only 5 are targeted to protrusions in all tested cell types, indicating that mRNA targeting is very cell type specific and that these 5 transcripts likely have conserved functional requirement in motile cells. Conserved and repeated sequence motifs were found in the 3'UTR region of these 5 transcripts, indicating a common targeting mechanism.

The authors then further focus on only 1 transcript, Rab13. Using an MS2-MCP system for visualization of the RAB13 mRNA distribution, in combination with several deletion versions of the RAB13 3'UTR showed the significance of the RNA motifs for localization of RAB13 mRNA to cellular protrusions and allowed identification of a minimal localization element crucial to exclusively polarize mRNA to motile EC protrusions and normal filopodia dynamics. Local translation of the polarized mRNA was shown using Puro-PLA assays.

Finally, the authors studied the function of rab13 mRNA polarization in ISV formation in zebrafish as an in vivo model of sprouting angiogenesis. Similarly, MS2-MCP visualization of rab13 distribution showed accumulation at the leading edge of the cell, but in this case the signal could be observed higher up in the filopodia. CRISP-Cas9-mediated excision of a part of the 3'UTR region confirmed that this region is crucial for polarization of the transcript to the leading edge, and results in miss-directed branching of the ISVs above the horizontal myoseptum.

Novelty:

The main finding of this paper is that local protein functions involved in cell and tissue movement can be directed to the correct subcellular location via mRNA-mediated compartmentalization. Polarization of mRNAs is known to be involved in cell migration, however, their precise functional roles remain largely unknown. Here, the authors were able to identify the cis-regulatory elements for Rab13, and showed that it's correct function in cell migration is dependent on correct targeting of its transcript and subsequent local translation.

General comments:

The methods used in this paper are appropriate, as well as the statistics used. Despite the fact that the authors found conserved sequence motifs in the 3'UTR of 5 transcripts that likely are all involved in conserved functions in motile cells, they only further investigated one of them. They hypothesize that a common targeting mechanism is involved in spatial localization of these transcripts. Confirmation of the importance of the localization element in at least one other gene would strengthen this hypothesis. Furthermore, identification of the trans-acting partner interacting with this recognition sequence and mediating the trafficking of Rab13 mRNA (and potentially all cluster k5 mRNAs) to their target location would further improve the informational value of this paper. The overall functional importance of their finding for guided tissue morphogenesis seems somewhat overstated as the phenotype in vivo appears not particularly convincing. Others have shown that guidance of ISVs in can occur in the complete absence of filopodia. Misguidance could therefore be the result of other effects that are not directly caused by ectopic filopodia. In the absence of more conclusive mechanistic data some of the conclusions appear premature or overstated. Having said that, this is a very interesting and original piece of work.

Specific comments:

- It seems Extended Data Table 1 is missing

- Legend figure 1d,i: meaning of arrows should be explained in the legends (now it is only explained in the Extended Data legend)
- The data in the paper would be easier to understand if the general mechanism or principle of some of the crucial methods used would be better explained, like the MS2-MCP system and Puro-PLA. The data are not easily comprehensible for non-experts.
- It is not clearly explained why only and specifically RAB13 was further investigated. Is the only reason the presence of 5 motif repeats in a short region of the 3'UTR? Using the MS2 system to test also the importance of the conserved RNA motifs for localization in another transcript would further strengthen the conclusion.
- Supplementary Video 1: it would help to better see the colocalization of the MCP-GFPnls signal and the position of filopodia if the arrowheads pointing to newly formed filopodia were also indicated on the left panel and/or if the position of the MCP-GFPnls signal was indicated in the right panel
- RAB13 siRNA knockdown results in increased filopodia formation. Fig3d seems to show more filopodia although quantification in Fig 3e and f indicates decreased number of filopodia
- Figure 3b,c: how come puro-PLA punctae are detected in the -Puro control?
- There is no reference to Figure 3d in the text
- rab13d3'UTR/d3'UTR embryos: why does the miss-directed branching only occur at the horizontal myoseptum and not upon sprouting from the dorsal aorta? Can the aberrant branching phenotype also be observed in other sites of sprouting angiogenesis, like hindbrain vasculature for instance?
- What is the effect of induced ISV hyperbranching, e.g. by Notch inhibition, on rab13 mRNA localization?
- Supplementary Video 2: it is hard to distinguish the MCP-GFPnls signal in this line and to look at the distribution of the signal over the cell. Why is the localization of the rab13 mRNA different than in vitro (in the filopodia rather than at the base)? Despite what is mentioned in the text, the cell in Video 2 is not a tip cell. It would be better to look at the rab13 mRNA distribution in true tip cell
- The region that was excised from the 3'UTR of the zebrafish rab13 transcript is quite big compared to the minimal localization element determined in HUVEC, this might affect also other regulatory mechanisms. Can the same RNA motifs and minimal LE be determined in the zebrafish 3'UTR region allowing more precise deletion?
- What is the trans-acting partner to mediate targeted localization of Rab13 mRNA and likely also the other cluster k5 members?
- It would be interesting to also find out the functional role of the other mRNAs with conserved polarization
- Is the effect of the LE deletion a real loss of function or rather a gain of function that causes "misguidance". What happens when overexpressing the deletion mutant in the background of WT Rab13mRNA? This may help to distinguish between the effects.

Referees' comments:

Referee #1 (Remarks to the Author):

The manuscript by Costa et al. reveals a functional role for subcellular localization of an endogenous mRNA (Rab13) in endothelial cells using elegant experiments (including gene editing of localization elements) in cell culture and in zebrafish. Although Rab13 mRNA was previously shown to localize in protrusions in cultured cells (by Mili and Nielsen and colleagues) the functional significance of this process was unclear. And while requirements for mRNA localization elements have previously been reported in other organisms, notably yeast and *Drosophila*, this study is the first such example in vertebrates. This is therefore an important study that is likely to be seen as a landmark in the mRNA localization field. The paper should also appeal more generally to those interested in cell migration and endothelial cell biology as the authors provide evidence that filopodia formation is guided by a local translation-based mechanism that prevents formation of these structures at ectopic sites (rather than an active mechanism to promote filopodia formation at one location). In addition to the advances summarized above, the authors also characterize other mRNA localization patterns in endothelial cells that pave the way for future work in this system.

I am enthusiastic about the manuscript but additional controls are required to back up some of the key conclusions. In addition, some clarification is needed to place the findings in the context of other studies on Rab13 mRNA localization.

We appreciate the Referee's interest and enthusiasm in our manuscript, clearly reflected in their positive comments. This is indeed the first study that addresses the role of mRNA localisation in the context of tissue formation in a vertebrate, with the potential to become a "landmark" study in the field. Moreover, the reviewer rightly mentions that by being the first to report on mRNA localisation in the control of filopodia formation and endothelial cell function, our study "paves the way" for further dissection of the critical roles of mRNA localisation in the general control of motile cell polarity, as well as in blood vessel growth.

Major points:

1. Rab13 appears to have different predicted splice isoforms (at least in human cells) and 3'UTR sequences can affect splice site selection (e.g. see <https://doi.org/10.1016/j.celrep.2019.05.083>). Defects in splicing could in principle account for the phenotypes observed, independently of mRNA localization, and it is therefore important that the authors check if patterns of splicing are altered by the deletion in the 3'UTR. The authors should also quantify Rab13 mRNA levels in wild-type and 3'UTR mutant HUVECs to strengthen their case that changes in overall mRNA levels do not contribute to the phenotypes in this system.

The Referee raises an important point that was not discussed in the previous version of our manuscript. To our knowledge, whilst the human *RAB13* has 5 splice variants resulting in 2 protein and 3 non-protein coding transcripts (Table I and Fig. I below), the zebrafish orthologue has 2 protein coding splice variants (Table II and Fig. II below). To address the Referee's concern regarding defects in mRNA splicing upon gene-editing, we performed new RNAseq experiments to interrogate *RAB13/rab13* splicing patterns in HUVECs and zebrafish embryos following CRISPR/Cas9-mediated genomic excision of the localisation elements (LEs). Our sequencing results (RNAseq map reads now included in Fig S4E and Fig S7E) show that edited and non-edited HUVECs/zebrafish embryos display equivalent RNAseq patterns, consistent with the presence of only one protein coding splice variant in both edited and non-edited cells (human *RAB13-201* and zebrafish *rab13-201*, Table I and II, Fig. I and II below). Thus, we now confirm that excision of the LE does not disrupt the splicing pattern of *RAB13/rab13*.

Name	Transcript ID	bp	Protein	Biotype	CCDS	UniProt	RefSeq Match	Flags
RAB13-201	ENST00000368575.5	1164	203aa	Protein coding	CCDS1058	P51153	NM_002870.5	TSL:1GENCODE basicAPPRIS P1MANE Select v0.8
RAB13-202	ENST00000462680.5	2123	No protein	Processed transcript	-	-	-	TSL:2
RAB13-203	ENST00000484297.1	643	No protein	Processed transcript	-	-	-	TSL:5
RAB13-204	ENST00000495720.5	868	No protein	Processed transcript	-	-	-	TSL:5
RAB13-205	ENST00000614713.4	1353	122aa	Protein coding	CCDS72921	A0A087WWB9	-	TSL:5GENCODE basic

Table I - List of human *RAB13* splice variants.

Figure I - Human RAB13 splice variants – ensemble genome browser data.

Name	Transcript ID	bp	Protein	Biotype	UniProt	Flags
rab13-201	ENSDART00000046922.5	1823	200aa	Protein coding	Q7T3A4	APPRIS P1
rab13-202	ENSDART00000131698.3	499	96aa	Protein coding	E9QI28	CDS 3' incomplete

Table II - List of zebrafish rab13 splice variants.

Figure II - Zebrafish rab13 splice variants – ensemble genome browser data.

The Referee also suggests that *RAB13* mRNA levels in CRISPR/Cas9-edited HUVECs should be quantified to confirm that observed phenotypes are not due to perturbation of mRNA levels. As such, we have added new data quantifying smFISH spot numbers to define absolute mRNA levels in both edited and non-edited HUVECs (Fig S4G) and observed no disruption to mRNA levels upon genomic excision of the localisation element (LE). Interestingly, correlation of *RAB13* mRNA spot counts versus the mRNA polarisation index revealed that polarisation of the mRNA is actually independent of mRNA levels, and is only perturbed upon excision of the LE (Fig S4H). Most importantly, loss of the LE did not alter total protein levels (Fig. S4I). Hence, we now confirm that phenotypes observed in gene-edited HUVECs are not a consequence of perturbed *RAB13* mRNA levels or translation, but are a consequence of the mis-localisation of *RAB13* mRNA.

2. The consistent phenotypic outcomes of removing the localization elements in the Rab13 3'UTR in HUVECs and zebrafish (which requires different gRNA sequences) argues against off-target effects of the CRISPR/Cas9 procedure being responsible for the observed phenotypes. Nonetheless, more needs to be done to strengthen this argument, which is the central part of the study. Do the gRNAs used in each experiment have predicted off-target effects? If so, the authors should check that potential off-target sites are not edited in the cells/animals used.

A powerful experiment, which I would class as desirable but not essential, would be to test if the defects caused by removal of the localization element make cells/animals particularly sensitive to partial knockdown of Rab13 (by incomplete siRNA or heterozygosity for a null mutation in the case of zebrafish). If this were the case, it would be very strong evidence for the specificity of the CRISPR/Cas9 phenotypes.

Or did the authors recover fish with wild-type Rab13 alleles that went through the CRISPR/Cas9 procedure? A strong off-target effect might be expected to give phenotypes when this wild-type Rab13 allele is homozygous.

The Referee raises a key point, as it is possible that gRNA pairs used in CRISPR/Cas9 gene editing could modify off-target sequences, albeit at a predicted very low frequency (CRISPRscan). Nevertheless, we have now performed RNAseq experiments using edited HUVECs/zebrafish embryos and carefully analysed the sequence

alignments corresponding to off-targets predicted by the gRNA designing tools. In contrast to the precise deletion of the LEs in the human and zebrafish *RAB13/rab13* 3'UTRs, we found no nucleotide mismatches in potential off-target sequences of expressed transcripts (now included in Fig S4F and Fig S7F), confirming the high-specificity of these tools.

The Referee also suggests several “desirable but not essential” approaches to further test the specificity of CRISPR/Cas9 phenotypes. One suggestion was to assess the phenotype of CRISPR/Cas9-derived zebrafish embryos that retain homozygous wild type *rab13* alleles. Indeed, our observations are based on experiments in line with the Referee’s suggestion. Using CRISPR/Cas9, we established a zebrafish line that carries a heterozygous edited *rab13* 3'UTR and these founders were incrossed to generate experimental embryos containing homozygous wild-type, heterozygous wild-type / edited and homozygous edited *rab13* 3'UTRs at Mendelian ratios. Hence, in the manuscript, homozygous mutant zebrafish embryos are consistently compared against wild-type siblings derived from the same clutch. Consequently, if the phenotype were associated with unidentified CRISPR/Cas9 off-targets effects, it would not robustly co-segregate with the *rab13* 3'UTR mutant genotype over multiple generations. However, phenotypes have consistently segregated according to *rab13* 3'UTR mutant genotype over all generations analysed to date, confirming the absence of CRISPR/Cas9 off-target effects.

Regarding testing of the specificity of CRISPR/Cas9 phenotypes in HUVECs, the Referee suggested further “desirable but not essential” experiments to determine the sensitivity of *RAB13* 3'UTR-edited cells to *RAB13* knockdown. A prediction here was that HUVECs lacking the *RAB13* 3'UTR LE would be more sensitive to partial loss of *RAB13*. However, in a parallel study by Moissoglu et. al. that is also under consideration at *EMBO Journal*, it was shown that knockdown of *RAB13* does not enhance the defects in cell motility / protrusiveness observed upon disruption of *RAB13* mRNA polarisation. This is due to the targeting of *RAB13* mRNA to the cell periphery being essential to the co-translational activation of RAB13 protein at this site, hence, *RAB13* knockdown does not further perturb RAB13 function at the cell cortex. Consequently, we would not expect an increased sensitivity of *RAB13* 3'UTR-edited HUVECs to additional knockdown of *RAB13*, consequently this precludes this as an approach to explore off-target effects. However, it should be reiterated that we did not observe any edits to predicted off-target sites in mutant HUVECs by RNAseq (Fig. S4F). Moreover, as the referee states, similar phenotypic outcomes in HUVECs and zebrafish “argues against off-target effects”. Finally, in the parallel study by Moissoglu et. al., a complimentary phenotype was observed in MDA-MB-231 cells using an alternative non-genetic approach involving antisense oligonucleotides. Hence, we can confirm that observed phenotypes are not due to any off-target effect of CRISPR/Cas9 editing, but due to mutation of the *RAB13* 3'UTR.

3. The importance of Rab13 mRNA localization for cell migration was previously challenged by the observation that APC is required for localization of this mRNA but not for cell migration (summarized in <http://www.jbc.org/cgi/doi/10.1074/jbc.R116.715193>). Costa et al. need to discuss this point. Is APC required for Rab13 mRNA localization in HUVEC cells? If so, might the requirements for APC-mediated Rab13 mRNA localization be specific for certain cell types? Is it known if APC is required for branching in the endothelial system in zebrafish?

The Referee raises an important point regarding the seeming discrepancy between (1) the well-established role of APC in targeting *RAB13* mRNA to the periphery of migrating cells and (2) observations that APC knockdown does not impact cell migration in certain cell types. As the referee indicates, this may suggest that APC-mediated *RAB13* mRNA localisation might not be important for cell migration. However, consistent with a role for APC in the regulation of RAB13-mediated endothelial cell migration, APC is indeed critical for HUVEC migration (Harris and Nelson, 2010) - although roles in zebrafish vascular development remain unexplored. More importantly, the role of *RAB13* mRNA localisation in the control of motile cell behaviour has been independently confirmed by a parallel study also under consideration at *EMBO J* (Moissoglu et. al.). Hence, our work, together with that of Moissoglu, et. al. clearly demonstrates that *RAB13* mRNA localisation is a conserved regulator of migration in diverse cellular contexts.

However, it is also clear that APC does not modulate cell migration in all cell contexts. For example, despite demonstrating important roles for *RAB13* mRNA localisation in MDA-MB-231 cell migration (Moissoglu, et. al., under consideration at *EMBO Journal*), Mili and colleagues have previously demonstrated that APC is not required for fibroblast migration (Mili, Moissoglu and Macara, 2008). A full explanation for this context-dependency to APC and RAB13 function have yet to fully defined, but likely relate to differences in the mode of motility used by distinct cell types, the type of dynamic protrusions employed (i.e. RAB13-dependent or not) or the extent to which distinct cell types rely on the spatial control of RAB13-mediated protrusiveness to orient cell movement. Indeed, exploration of these scenarios is a focus of future work in the laboratory. Further discussion of these points can be added to the manuscript text, if requested.

4. Additional controls are needed for the PURO-PLA experiments. The authors should repeat the experiment when Rab13 is knocked down by siRNA in order to confirm the specificity of the antibody in these experiments

(not just for western blotting). It would also be desirable to repeat this analysis in the cells that lack the region of the 3'UTR required for mRNA localization as this will allow a direct assessment of the link between mRNA localization and local translation.

We agree with the Referee that these data would further strengthen observations that *RAB13* is locally translated at protrusions. However, we would like to point out that these observations were recently independently validated by (Moissoglu *et al.*, 2019). Using similar Puro-PLA tools, as well as the SunTag live translation reporter system, Moissoglu *et. al.* confirmed that *RAB13* mRNA is locally translated when targeted to protrusions. Hence, although we did not have time to acquire the requested data prior to the COVID-19-related closure of research facilities, the direct “link between mRNA localization and local translation” of *RAB13* has already been fully validated.

Importantly, this recent work from Moissoglu *et. al.* significantly adds to the impact of our study. A core message of our manuscript is that the local targeting of *RAB13* mRNA functions to spatially compartmentalise RAB13 protein action, which in turn orients motile cells both *in vitro* and *in vivo* during tissue morphogenesis. One way to compartmentalise RAB13 activity would be if such spatial targeting of mRNA could alter the site of RAB13 protein synthesis. This was briefly touched upon in our original submission following inclusion of Puro-PLA data demonstrating that *RAB13* mRNA could be locally translated at protrusions. However, Moissoglu *et. al.* now fully confirms that this is the case using several approaches, providing strong support for our core observation that spatial control of *RAB13* mRNA localisation can define the site of RAB13 protein action. Indeed, in a parallel study also under consideration at *EMBO Journal*, Moissoglu and colleagues further reveal that RAB13 can co-translationally assemble with its GEF, RABIF, to locally activate protein. Hence, the work of Moissoglu *et. al.* beautifully complements our study by defining a solid mechanistic model for understanding how local targeting of *RAB13* mRNA can achieve spatial control of RAB13 activity, as described by our work, which ultimately serves to orient motile cell polarity and tissue movement.

Minor points:

1. How conserved is the region of the Rab13 3'UTR that mediates localization? An alignment of Rab13 sequences from different species (including human and zebrafish) should be provided in the Extended Data section.

As requested by the Referee, we have now performed a multi-species sequence alignment of the *RAB13* LE and added it to the revised manuscript as Fig. S6F. Interestingly, despite the highly conserved roles for both human and zebrafish 3'UTRs in regulating *RAB13* mRNA localisation, we observed a relatively low frequency of ~40% absolute nucleotide similarity across the several *RAB13* 3'UTRs analysed.

2. The type of statistical tests used for each panel should be specified in the legends or a supplementary table (depending in the journal's policy). Currently, the methods state that different tests were performed depending on the data distribution, which is correct, but the reader needs to be able to understand which test was applied where.

We agree with the Referee that the specific statistical tests should be mentioned and details of each test have been added to the respective figure legends.

3. The manuscript is mostly clearly written but in some sections it can be made much more concise. There are also several typos and spelling mistakes in the manuscript:
E.g.: miss-localization, longstading.

We thank the Reviewer for pointing out mistakes in the manuscript text and we have corrected them accordingly.

4. In the abstract, insert 'mRNA' between 'endogenous RAB13' and 'targeting' and revise 'endogenous gene-edited mRNAs', which is a potentially misleading term.

We thank the Reviewer for the suggestions. mRNA has been inserted between “*RAB13*” and “targeting”; “endogenous gene-edited mRNAs” has been replaced by “gene-editing”.

5. Page 5: “exclusive targeting of mRNA to sites of incipient filopodia formation in cellular protrusions”. Is targeting really exclusive? Also, is the mRNA in cellular protrusions or in a broader region of the cell that is prone to form cellular protrusions? This is an important distinction when trying to understand the spatio-temporal aspects of the system.

We agree with the Referee that “exclusive targeting” may be a misleading term. Moreover, we apologise for being unclear regarding the localisation of *RAB13* mRNA. As such, we have clarified this text by replacing it with “enriched targeting of mRNA to filopodial protrusions, as well as broader sites of incipient filopodia formation”.

6. Mili et al. recently reported on the relationship between Rab13 mRNA localization and translation (<https://elifesciences.org/articles/44752>). Although this work was in different cell lines to those used by Costa et al. the authors should consider citing relevant findings from this study.

This work had not been published at the time of submission. As such, we thank the Reviewer for the suggestion and now refer to this work in our new manuscript draft.

7. Extended data Figure 2a: is the motif predicted to form a G-quadruplex?

The reviewer raises an interesting question, given that quadruplexes have been implicated in crucial aspects of mRNA biology, including targeting (Kharel *et al.*, 2020; Subramanian *et al.*, 2011). We have used the *Quadruplex forming G-Rich Sequences* (QGRS) mapper (<http://bioinformatics.ramapo.edu/QGRS/index.php>) to analyse the motif and indeed detect a predicted quadruplex. The results of this analysis are displayed in Table III below. However, whether this motif truly forms a quadruplex *in vivo* and, if so, whether this structure is responsible for recognition by APC and mRNA transport remain to be explored. These questions will certainly be the focus of follow-up studies, but we feel that they are beyond the scope of the current manuscript.

Position	Length	QGRS	G-Score
4	16	GGUAAAAGGAGGAAGG	17

Table III - QGRS analysis of the motif AAGGGUAAAAGGAGGAAGGGA.

Referee #3 (Remarks to the Author):

In this manuscript, the authors investigate how polarized localization of mRNAs can contribute to tissue morphogenesis. Through RNAseq analysis of endothelial cell bodies and protrusions, they identify transcripts that are selectively localized to cellular protrusions. Comparison with other cell types from published studies identified a core set of protrusion-localized genes, including RAB13, which they investigated further. They identify a conserved cis element in the 3' UTRs of these transcripts that suggested it is responsible for compartmentalization. Deletion of this element in RAB13 in human cells disrupts its localization and leads to defects in filopodial dynamics. Further investigation in zebrafish suggests this UTR element is conserved, required for cell compartmentalization, and is important for directional migration during endothelial sprouting.

Conceptually, this work is a potentially impactful move forward in functionally linking the significance of subcellular mRNA localization with a cellular and tissue morphogenetic role. However, there are significant experimental and technical gaps that leave the work incomplete. Most importantly, any data on how mRNA localization (and lack thereof) actually affects cell and tissue morphogenesis is completely lacking. Therefore, the studies as they currently stand are only suggestive and preliminary. Particular details are listed below.

We appreciate the Referee's recognition that our work is impactful and will move the field of mRNA localisation forward, but also acknowledge that we only briefly touch upon the mechanistic link between mRNA localisation and downstream changes to cell behaviour. However, it is important to highlight a parallel manuscript by Moissoglu et. al., which is also currently under consideration at *EMBO Journal* and perfectly complements our work by filling in these mechanistic gaps. In particular, whilst we show that disruption to the polarised distribution of RAB13 mRNA alters the spatial pattern of actin remodelling in endothelial cells and perturbs tissue movement, Moissoglu and colleagues demonstrate that RAB13 mRNA targeting can achieve this by dictating the location of RAB13 activity via control of its co-translational interaction with its GEF, RABIF. Hence, the work of Moissoglu et. al. greatly strengthens our novel findings by providing a solid mechanistic basis for why RAB13 mRNA targeting can influence the location of filopodia remodelling, motile cell polarity and oriented tissue movement, by defining the sites of local RAB13 GTPase activation.

Major concerns.

The computational work applied throughout the study is incompletely described and a bit weak. For example, RNAseq analysis of HUVEC protrusions only relies on duplicate samples and no statistical values are given (e.g. in Fig 1b).

We understand the Referee's concerns regarding a lack of reference to statistical values. As such, we have now applied a statistical cut-off to the data based on the false discovery rate (FDR) p-value (q-value). In the latest version of the manuscript, the RNAseq data is now plotted in the form of a volcano plot that displays protrusion/cell body enrichment against FDR (See Fig. 1B). mRNAs with a fold enrichment >1.6 and $q < 0.05$ are considered to be enriched in protrusions over cell bodies and are now represented as green dots in the volcano plot.

This issue is further compounded by the k-means clustering, which shows a pretty clear cluster for the previously published data, but is much less convincing for HUVEC. It seems the authors could have easily identified this signature without the HUVEC analysis. If they insist on including the HUVEC data, appropriate statistical cut-offs should be applied, as well as any manipulations to apply normalization or batch effects to accurately compare the public data with their own.

To clarify, selection of mRNAs for clustering analyses is fundamentally restricted to only transcripts enriched in HUVEC protrusions. If clustering is performed without these transcript restrictions, the analyses generates clusters containing mRNAs not even expressed in this cell type and/or not enriched in HUVEC protrusions. As such, the HUVEC protrusion data is essential, as k-means clustering analyses lacking this data would not reveal the cluster of potential universal modulators of motile cell behaviour that we identify. However, we agree with the Referee that a statistical cut-off must be applied and the data must be normalised. Thus, we have now selected HUVEC protrusion-enriched mRNAs based not only on a fold enrichment threshold >1.6 but also the statistical cut-off $q < 0.05$. Moreover, to normalise the data, the protrusion/cell body enrichment levels of the public RNAseq data sets from other cell lines that we used were extracted, scaled and centred, prior to inclusion in k-means clustering analyses. Although the new clustering output is inevitably slightly different to the one presented in the original manuscript submission, mRNAs identified in k5 consistently remain exclusively clustered together (Fig. 1C, S1A).

Similarly, it is unclear how the cis regulatory motif was identified. Was this by computational means? Is this significantly enriched in only the k5 protrusion-localized transcripts?

The cis-regulatory motif was identified using the MEME Suite (<http://meme-suite.org/>) and information on the motif is now provided in Table S3. As requested, we have also included an additional panel to Fig S2B demonstrating that the motif is present in 100% of the mRNAs clustered in *k5*, but at a much lower frequency in all remaining clusters, as determined by FIMO motif analyses.

What happens if they take a randomly selected set of transcripts and search for over-represented motifs? Presumably, their motif would be more likely to be found in other protrusion-localized transcripts, regardless of cell type - is this the case? What about conservation in other species?

The Referee raises an interesting question about the specificity of the identified mRNA motif to only protrusion-localised transcripts, and about conservation of this motif between species. To address these points, first we took the 3'UTRs of the 15-most cell body-enriched transcripts in our RNAseq analyses (i.e. non-protrusion targeted mRNAs) and performed a motif discovery search using the MEME Suite. As expected, the enriched motifs identified by this search bear no resemblance to the *k5* cluster motif identified in our study (See Fig. IV below). When combined with the evidence that this *k5* cluster motif is underrepresented in transcripts from other mRNA clusters with distinct polarisation patterns (See answer to comment above), these data are consistent with a high specificity of this motif to *k5* cluster mRNAs and a key role in the targeting of these mRNAs to protrusions.

Second, to investigate conservation of the *k5* mRNA motif between species, we scanned the 3'UTR of various *RAB13* orthologues for the presence of this motif. We found the motif to be present in 3'UTRs of all tested mammalian orthologues, but observed little conservation through to the zebrafish *rab13* 3'UTR, which may be expected considering the evolutionary distance between the human and zebrafish genomes. Importantly, the homologous motif we identified in the mouse transcript precisely overlapped the region of the *Rab13* 3'UTR shown to promote mRNA targeting in NIH/3T3 fibroblasts in a parallel study by Moissoglu et. al., which is also under consideration at *EMBO Journal*. Hence, at least in mammals, the *RAB13* mRNA motif is highly conserved between species at both the level of sequence and function.

Figure IV - Motifs over-represented in 3'UTRs of mRNAs enriched in cell bodies.

Regarding the cis regulatory motif, the authors would greatly strengthen their study if they could more rigorously identify a minimal motif that was sufficient to cause localization (even if it needs to be in multiple copies). In this regard, finer deletion analysis/point mutations and heterologous assays would be welcome. This would allow for better assessment of its occurrence in other transcripts. Further, it may provide additional insights on mechanisms that contribute to its localization.

We agree with the Referee that finer mutational analysis of the *k5* mRNA motif could potentially identify a minimal motif responsible for mRNA targeting to cell protrusions. However, from both existing data and new data we have acquired, it appears that distinct gene transcripts have very different minimal requirements for the number of motifs needed to drive targeting. For example, our previous deletion analysis of four motifs in the *RAB13* 3'UTR using the MS2 system (Fig. 1J) revealed that deletion of each motif individually was insufficient to block targeting. In contrast, a larger deletion of all four 3'UTR motifs was sufficient to entirely disrupt *RAB13* targeting. Likewise, constructs containing only two of the *RAB13* 3'UTR motifs were unable to drive targeting to protrusions, whereas those with four motifs were. Hence, *RAB13* has a minimal requirement of three to four motifs for mRNA targeting. However, our new data reveal that protrusion targeting of another cluster *k5* mRNA, *TRAK2*, can be entirely disrupted upon precise deletion of just one motif (See Fig. S2E). These new data further confirm the critical role of the 3'UTR motif in *k5* mRNA targeting, but they also clearly demonstrate that distinct transcripts require different numbers of this motif. For this reason, we believe that a universal minimal motif for mRNA polarisation may not exist, and the full context of the LE, in which several copies of the motif and other key regulatory sequences may be embedded, is required for robust mRNA targeting and varies between genes.

In the absence of a consequential cellular or tissue morphogenesis phenotype for *RAB13* (both of which are missing so far here), it may require disruption of the localization mechanism itself (i.e. whatever protein is binding to this motif) to perturb multiple effectors and see a phenotypic effect.

Here, the Referee suggests that a more acute *in vivo* phenotype may be observed upon combinatorial disruption of the trafficking of all *k5* mRNAs. This is a good point, yet it is key to point out that the objective of this *in vivo* work was not to obtain the most severe vascular phenotype, but to corroborate *in vitro* observations that *RAB13* mRNA targeting functions to orient motile cell polarity in an *in vivo* context. Prior to our study, a vast body of literature had investigated polarised mRNA targeting during cell migration *in vitro*, but the function of this phenomenon and any *in vivo* relevance were lacking. Whilst observation of a highly acute phenotype upon the mutation of the *rab13* LE would have been compelling, our work is the first to reveal a clear conserved role for mRNA targeting in the spatial regulation of motile cell polarity, both *in vitro* and more importantly during oriented tissue movement in a vertebrate organism. Hence, this study represents a step-change in our understanding of the general function and relevance of this phenomenon to tissue formation. Indeed, it is remarkable that we observed such an obvious phenotype *in vivo* at all, considering that it is only the location of a single mRNA, not its expression, that is perturbed in mutant embryos.

Nevertheless, the Referee correctly points out that perturbation of the global localisation mechanisms that transport *RAB13* and other *k5* cluster mRNAs may generate a more acute phenotype. It is well established that polarised trafficking of *RAB13* mRNA is mediated by APC in multiple cellular contexts (Mili, Moissoglu and Macara, 2008; Wang *et al.*, 2017). However, in addition to mRNA trafficking, APC also plays key roles in the regulation of Wnt/ β -catenin signalling pathway and consequently zebrafish *apc* mutation induces severe pleiotropic, early embryonic lethality phenotypes (Hurlstone *et al.*, 2003). As such, dissecting the specific role of APC-mediated mRNA trafficking in this mutant background would be highly challenging. As an alternative, considering our new data defining the functional targeting regions of all *k5* mRNAs (See Fig. S2D,E), we could generate combinatorial zebrafish LE mutants for all of the *k5* cluster mRNAs. However, this would require a considerable effort (~9 months to 1 year), and as the priority of this study was to focus on *RAB13/rab13* mRNA function, we feel these experiments are beyond the scope of the current manuscript and will be a focus of future studies.

For the genome editing in HUVEC, I was always under the impression that isolating HUVEC clones was very challenging given that they are primary cells and will senesce over a limited number of passages. This raises some concerns about these experiments, especially since no controls were performed (e.g. parallel generation of clones using irrelevant sgRNA, or targeting downstream of the UTR, also a CDS mutation is important for comparison). Multiple clones would also be important. All of these particular controls are important to make sure the observed results are not arising due to some other issue with deriving HUVEC clones. It seems that this core set of polarization-localized genes is in many cell types. Perhaps it would be easier to pick the cell type (or better yet multiple types) that is most amenable to manipulation for these studies.

The Referee's suggestion of controls for the CRISPR/Cas9 editing of HUVECs is extremely important and we apologise for a lack of clarity regarding our methodology, as we had already performed such controls. First, as the Referee states, HUVECs do indeed have a limited passage span, hence, HUVECs were not used beyond passage 7 and did not display senescence prior to use. Second, we did indeed include appropriate controls as requested by the Referee. Control cells (labelled as wild-type cells in the manuscript) were clonally derived from individual HUVECs that expressed co-transfected GFP plasmid but showed no CRISPR-Cas9 excision of the LE in the 3'UTR. These cells were always expanded and tested in parallel with mutant cells to avoid issues related to clonal derivation of HUVECs. As such, these cells were the ideal control as they were treated with the same sgRNA and were processed under the exact same conditions as mutant cells, but just did not acquire the *RAB13* mutation. Finally, as requested by the Referee, all the data were indeed collected from multiple clones (3x independent wild type clones and 3x independent mutant clones). To clarify these points, we now add these details to the respective Figure legends and Methods text, as well as include an extra figure panel detailing this methodology (Fig S4A).

The HUVEC studies lack any assessment of migratory or protrusive activity. Are these affected by disrupting *RAB13* localization? What about directionality? Persistence? Can they sprout in 3D matrix? Form tubes in matrigel? Etc. . . ? The authors need to assess if there is actually an effect on cell migratory behaviors, otherwise the relevance of their current findings is unclear.

We agree with the Referee that, although we do demonstrate that *RAB13* mRNA targeting achieves spatial control of membrane protrusions *in vitro* and oriented movement *in vivo*, further characterisation of effects on cell migration *in vitro* would be useful. As such, we now point the reviewer to a parallel *in vitro* study by Moissoglu *et al.* that is also under consideration at *EMBO Journal*, and which defines key roles for *RAB13* mRNA localisation in the control of cell protrusion/retraction, migration and invasion through Matrigel. As such, the study by Moissoglu *et al.* perfectly complements our work by revealing conserved roles for *RAB13* mRNA targeting in cell migration. Moreover, where Moissoglu and colleagues fail to address questions related to why migrating cells would choose to polarise *RAB13* mRNA at precise focal sites within cells or any *in vivo* relevance of this phenomenon, our work fills in these gaps by uniquely demonstrating that this mRNA polarisation functions to spatially orient cells both *in vitro* and *in vivo*. Hence, taken together, these complementary observations

significantly enhance the impact of both studies and provide a complete answer to Referee's question.

The zebrafish studies suffer from some of the same issues. The authors need controls for the MS2 localization studies - e.g. a construct with a mutated or deleted LE motif and/or one with "irrelevant" UTR for comparison. It is also important to generate a coding sequence mutation for comparison to the UTR deletion. The authors should provide an alignment of the cis motif in fish and humans (other species would be good to see as well) to assess conservation. This may also aid in finding a minimal sufficient localization motif, as noted above.

The Referee raises several valid points regarding our studies in zebrafish, which we have now addressed with new data. First, as requested by the Referee, we have added new data showing that control constructs lacking the *rab13* LE fail to promote transcript targeting to the leading edge of migrating cells *in vivo* (new Fig. S6D). Hence, co-targeting of the *rab13* transcript and MCP-GFP is confirmed to be a specific property of the *rab13* LE.

Second, as suggested by the Referee, we have now investigated the effects of *RAB13* loss of function using CRISPRi (See Fig. III above). We chose this approach as mutation of many genes in zebrafish does not reveal full null phenotypes due to genetic compensation, whereas CRISPRi inhibits transcript elongation and is known to circumnavigate these issues (El-Brolosy *et al.*, 2019; Rossi *et al.*, 2015). Disruption of *rab13* using CRISPRi generated a very similar phenotype to previous knockdown studies using morpholino oligonucleotides (MOs) (Wu *et al.*, 2011), whereby ISVs appeared heterogenous in length and were frequently stunted versus *cav-1* gRNA-injected controls (Arrowheads in Fig. IIIB above). This phenotype is more severe than that observed upon mutation of just the *rab13* LE, but as discussed in response to Referee 2, this is expected as all Rab13-mediated cellular functions are perturbed by CRISPRi/MO knockdown, whereas only local activation of Rab13 at the cell periphery is disrupted in *rab13* LE mutants. As such, the CRISPRi/MO phenotypes aid understanding of the global function of Rab13, but provide minimal insights into the specific function of *rab13* mRNA targeting. In contrast, phenotypes observed upon *rab13* LE mutation can be specifically ascribed to loss of *rab13* mRNA polarisation.

Finally, as request by the Referee, we have now added a sequence alignment of the human *RAB13* LE with other species (Fig. S6F). As discussed in response to Referee 1 and 2, this analysis reveals relatively poor conservation between the human LE and zebrafish (~57%; see Table IV above), despite full conservation of the targeting function of this region between species. Consequently, this alignment does not aid in identification of a universal minimal targeting motif, and indeed, the numbers of these motifs required to drive mRNA targeting may vary between distinct transcripts and species, as discussed above in response to another comment by Referee 3.

Along similar lines, the phenotypic characterization of the zebrafish mutant is quite preliminary. Numerous questions and issues need to be addressed: Where and when is *rab13* expressed? Lots of different non-autonomous defects could lead to the observed phenotype. Therefore it is important to know what the overall phenotype of the embryo is. For the ISV defects, is there a defect in circulatory system patterning or function? How do defects compare to a full deletion of the coding sequence or a UTR deletion that does not remove the cis element? Regardless of cell type affected, experiments to address cell autonomy must be performed, eg. transplants or transgene-directed rescues. The latter approach on a complete *rab13* deletion allele would enable cell autonomous rescue with different UTR constructs and should be considered. I would note that if the goal of their efforts is to demonstrate that mRNA localization is important for tissue morphogenesis, then what the actual cell type is that exhibits a defect is inconsequential - but all of the aforementioned issues need to be addressed regardless.

The Referee raises several valid points related to the phenotypic characterisation of *rab13* LE mutant embryos. First, using the single cell atlas recently released by the Miller lab (Farnsworth, Saunders and Miller, 2020) we reveal that *rab13* is expressed in a wide range of zebrafish embryonic cell types through day 1 to day 5 h.p.f (See Fig. V below). Importantly, *rab13* is highly enriched in the endothelial cell cluster that co-expresses *kdr* and *kdrl*, confirming expression in the vasculature. Moreover, our new smFISH data revealing *rab13* expression in explanted zebrafish endothelial cells further provides direct evidence that *rab13* is expressed in this cell population (Fig. 4B-D).

Next, we apologise for a lack of clarity regarding the overall embryonic phenotype *rab13* LE mutants. To confirm, embryos lacking the *rab13* LE are viable, can be recovered at classic Mendelian ratios and have no detectable additional embryonic phenotypes other than the observed defects in ISV branching. Moreover, there are no detected broader defects in cardiac function and/or gross vascular patterning that may underlie the observed path finding defects, indicating that this phenotype is highly specific to migrating endothelial cells. We have added extra text to the manuscript to clarify these points.

For a full discussion of how the *rab13* LE mutant phenotype compares to a global loss of *rab13* function, please see the response to the Referee's comment above this one. In summary, global loss of *rab13* function results in a more severe ISV stalling phenotype, indicating perturbation of more aspects of Rab13 function than observed

upon mutation of the *rab13* LE alone. As such, our work nicely dissects the key functional differences between peripheral targeted/translated *rab13* and the rest of the Rab13 protein pool.

Figure V - Single-cell RNA-seq atlas shows *rab13* and *kdrl* expression levels (Range) in a distinct endothelial cell clusters from 1 to 5 h.p.f.

Finally, the Referee suggests two alternative approaches to exploring cell-autonomy *in vivo*, but unfortunately both were impracticable in the context of this project. Firstly, cell transplantation is exceedingly challenging for such live-cell imaging studies with mutant strains. Only one in four donors will be mutant, limiting acquisition of live-cell imaging data to a maximum of one or two mutant-transplanted host embryos per week. Only hosts containing transplanted cells at the leading tip position of ISVs can be quantified (~25-50% hosts). Moreover, only ~40% of these cells will display defects in pathfinding. Consequently, completion of this work with sufficient numbers would require several months of effort exclusively dedicated to this experiment, which was not possible prior to the COVID-19-related closure of research facilities. Secondly, as for other Rab proteins, the extreme dose-sensitivity of cells to *rab13* expression precludes transgene-directed rescue experiments. Even subtle overexpression of wild-type *RAB13* is sufficient to disrupt membrane trafficking (Nokes, 2008), induce autophagy (Zhang, 2017) and disrupt cell-cell junction dynamics (Marzesco, 2002) amongst other processes. As such, the output of transgene-directed *rab13* expression cannot be clearly ascribed to gene rescue.

However, it is important to note that the design of our genetic approach already confirms the specificity of the observed phenotype, as *rab13* LE mutation specifically impacts only motile cells. For classic coding sequence mutations, transplantation / rescue experiments are useful to dissect cell-autonomous phenotypes, as all cells in the organism are affected by gene loss or gain of function. In contrast, the non-coding *rab13* LE mutation does not impact global mRNA or protein levels, but very precisely impacts *RAB13* activity just at motile cell processes and not at any other subcellular sites (Fig. S4G-I, S7J and parallel study by Moissoglu et. al. that is also under consideration at *EMBO Journal*). Hence, the *rab13* LE mutation exclusively affects only motile cells, emphasising the high specificity of this approach. Indeed, even within migrating endothelial cell populations, it is only the most motile of cells that require the *RAB13* LE for mRNA targeting and would be affected by LE mutation (See Fig. S3). As none of the tissues surrounding ISVs are motile at the experimental stages investigated, they would be unaffected by *rab13* LE mutation. Hence, observed phenotypes are conclusively a consequence of specific defects in endothelial cell movement.

Minor comments

Localization in the zebrafish. The smFISH experiments relied on explanted cells. Presumably sensitivity is an issue? Also, what is the localization of *rab13* protein in zebrafish? Obviously this is not trivial to determine in zebrafish, but would be important confirmatory data. For MS2-visualized localization, what is the pattern in non-tip cells?

The Referee raises several questions regarding *rab13* mRNA and Rab13 protein localisation in zebrafish endothelial cells. First, we can confirm that the broad expression of *rab13* mRNA in non-endothelial tissues precludes whole-mount smFISH assays to define mRNA localisation, as we cannot distinguish endothelial *rab13* from any transcript expressed in adjacent cells. However, rather than using explanted cells of unclear identity to

investigate *rab13* mRNA localisation, we have now added new smFISH experiments using explanted zebrafish endothelial cells (Fig. 4B-D). These assays clearly reveal polarised *rab13* mRNA in migrating endothelial cells, as well as disruption of this localisation upon mutation of the *rab13* LE.

Regarding protein localisation, we have been unable to robustly detect Rab13 protein in this setting (or in zebrafish *in vivo*), likely due to a lack of epitope recognition by antibodies raised against mammalian RAB13. This is a common issue with detection of zebrafish protein using pre-existing antibodies, and to rectify this issue we would need to raise a novel antibody using an equivalent zebrafish epitope. Nevertheless, studies in HUVECs (new Fig. S5E) and MDA-MB-231 cells (Moissoglu et. al., also under consideration at *EMBO Journal*) confirm that RAB13 protein is homogeneously distributed throughout the cytoplasm with a slight enrichment at perinuclear sites. Hence, the peripheral localisation of *RAB13* mRNA does not promote polarised localisation of RAB13 protein. In contrast, it serves to drive local *activation* of a subset of the protein pool at the cell periphery, as discussed in greater detail in response to several of the Referee's comments above, as well as in the parallel manuscript by Moissoglu et. al.

Finally, as the Referee requests, we have now added data using the MS2 system that highlights the localisation of *rab13* mRNA in non-tip cells (i.e. migrating stalk cells; See Fig. S6C and Movie S4). This new data reveals that *rab13* mRNA remains consistently polarised at puncta towards the front of these collectively migrating cells, although the functional relevance of this localisation in stalk cells remains to be investigated.

". . .monitored the dynamics of *rab13* 3'UTR mRNA. . ." is written as if you are monitoring endogenous transcript, but this is really the 3'UTR from a transgene. Should probably modify this to be more explicit.

We agree with the Referee that this sentence may be misleading and have modified it to "...monitored the dynamics of **the** *rab13* 3'UTR **reporter**..."

Referee #4 (Remarks to the Author):

Summary:

Using an RNAseq screen the authors identified mRNAs enriched in protrusions of migrating HUVECs. Of 320 identified transcripts, only 5 are targeted to protrusions in all tested cell types, indicating that mRNA targeting is very cell type specific and that these 5 transcripts likely have conserved functional requirement in motile cells. Conserved and repeated sequence motifs were found in the 3'UTR region of these 5 transcripts, indicating a common targeting mechanism.

The authors then further focus on only 1 transcript, Rab13. Using an MS2-MCP system for visualization of the RAB13 mRNA distribution, in combination with several deletion versions of the RAB13 3'UTR showed the significance of the RNA motifs for localization of RAB13 mRNA to cellular protrusions and allowed identification of a minimal localization element crucial to exclusively polarize mRNA to motile EC protrusions and normal filopodia dynamics. Local translation of the polarized mRNA was shown using Puro-PLA assays.

Finally, the authors studied the function of rab13 mRNA polarization in ISV formation in zebrafish as an in vivo model of sprouting angiogenesis. Similarly, MS2-MCP visualization of rab13 distribution showed accumulation at the leading edge of the cell, but in this case the signal could be observed higher up in the filopodia. CRISP-Cas9-mediated excision of a part of the 3'UTR region confirmed that this region is crucial for polarization of the transcript to the leading edge, and results in miss-directed branching of the ISVs above the horizontal myoseptum.

Novelty:

The main finding of this paper is that local protein functions involved in cell and tissue movement can be directed to the correct subcellular location via mRNA-mediated compartmentalization. Polarization of mRNAs is known to be involved in cell migration, however, their precise functional roles remain largely unknown. Here, the authors were able to identify the cis-regulatory elements for Rab13, and showed that it's correct function in cell migration is dependent on correct targeting of its transcript and subsequent local translation.

General comments:

The methods used in this paper are appropriate, as well as the statistics used. Despite the fact that the authors found conserved sequence motifs in the 3'UTR of 5 transcripts that likely are all involved in conserved functions in motile cells, they only further investigated one of them. They hypothesize that a common targeting mechanism is involved in spatial localization of these transcripts. Confirmation of the importance of the localization element in at least one other gene would strengthen this hypothesis.

Furthermore, identification of the trans-acting partner interacting with this recognition sequence and mediating the trafficking of Rab13 mRNA (and potentially all cluster k5 mRNAs) to their target location would further improve the informational value of this paper.

The overall functional importance of their finding for guided tissue morphogenesis seems somewhat overstated as the phenotype in vivo appears not particularly convincing. Others have shown that guidance of ISVs can occur in the complete absence of filopodia. Misguidance could therefore be the result of other effects that are not directly caused by ectopic filopodia. In the absence of more conclusive mechanistic data some of the conclusions appear premature or overstated. Having said that, this is a very interesting and original piece of work.

We appreciate the Referee's enthusiasm about the novelty of our findings and positive comments regarding our manuscript's experimental approach. Indeed, this is the first study to support the role of mRNA localisation in the spatial control of cell migration and most importantly, in tissue morphogenesis. For this, we generated the first tissue-specific reporter of mRNA localisation in a vertebrate animal, and use genetic techniques to uniquely manipulate a 3'UTR LE in zebrafish, an approach that has been rarely attempted *in vitro*.

However, the Referee does raise some valid concerns in their opening statement. First, the Referee is correct to mention that confirmation of the importance of LE function in other cluster k5 mRNAs and an understanding of the identity of trans-acting partners responsible for mRNA trafficking would be important. Indeed, in response to specific Referee comments below related to these issues, we now present new data confirming LE function in other k5 transcripts and provide discussion of the key role APC plays in RAB13/k5 mRNA targeting. In addition, the Referee cautions that work from another group, which demonstrates that complete disruption of filopodia does not promote mis-guidance of ISVs (Phng, Stanchi and Gerhardt, 2013), may contradict our observations. However, it is critical to note that the results of the study by Phng and colleagues are entirely consistent with our work. Phng et. al. revealed that loss of filopodia significantly delay ISV migration, and that tip cells lacking filopodia consistently stall at the horizontal myoseptum (HM). Hence, similar to rab13 LE mutants, ECs lacking filopodia struggle with this key pathfinding decisions at the HM location. Importantly, the complete disruption of filopodia explored by Phng and colleagues cannot be directly compared with the mis-localisation of filopodia we report. According to our model, cells lacking filopodia would not make incorrect guidance decisions as observed in rab13 LE mutants, but would take longer to make these decisions, consistent with observations by Phng et. al. In contrast, the ectopic expansion of filopodia we observe would promote promiscuous exploration of the local environment, mis-interpretation of surrounding guidance cues and subsequent poor pathfinding decisions. Hence, data presented by Phng et. al. is highly consistent with our work, as further discussed in response to specific Referee comments below. Moreover, a parallel study by Moissoglu et. al., which is also under consideration at *EMBO Journal*, reveals that the role of RAB13 mRNA targeting in modulating cell protrusions is

highly conserved between distinct cell types. Hence, roles for *RAB13* mRNA polarisation in the spatial control of membrane process dynamics and motile cell polarity may be a conserved feature of tissue formation in other cellular contexts.

Specific comments:

- It seems Extended Data Table 1 is missing

We apologise for this mistake and have included Extended Data Table 1 in this new submission.

- Legend figure 1d,i: meaning of arrows should be explained in the legends (now it is only explained in the Extended Data legend)

We thank the Referee for the suggestion and a description of the meaning of these arrows has now been added to the figure legends.

- The data in the paper would be easier to understand if the general mechanism or principle of some of the crucial methods used would be better explained, like the MS2-MCP system and Puro-PLA. The data are not easily comprehensible for non-experts.

The Referee indicates that some technical details are not fully detailed in the manuscript and should be made clear for the broad readership of the journal. To complement the diagrams that depict the principles underpinning MS2-MCP and Puro-PLA assays (See Fig S2C and Fig 3A, respectively), we have added new explanatory details to the manuscript text when introducing data based on these methods.

- It is not clearly explained why only and specifically *RAB13* was further investigated. Is the only reason the presence of 5 motif repeats in a short region of the 3'UTR? Using the MS2 system to test also the importance of the conserved RNA motifs for localization in another transcript would further strengthen the conclusion.

As the Referee hints, one reason why we prioritised the *RAB13* 3'UTR for functional interrogation was that it contains a group of 5 motifs within a short stretch that aided mutational analyses. In addition, as mentioned in the manuscript text, the *RAB13* protein also has strong links to regulation of cortical actin dynamics, making it an ideal candidate for spatial control of motile cell behaviour.

Regarding the Referee's second point, we agree that determining whether the *k5* mRNA motif is functional in transcripts other than *RAB13* is critical to prove that this conserved sequence underpins co-regulation of *k5* mRNA targeting. To address this point, we first generated MS2 constructs containing the 3'UTRs of *KIF1C* / *NET1* / *RASSF3* / *TRAK2* and confirmed that, similar to the *RAB13* 3'UTR, these sequences drive mRNA localisation to endothelial cell protrusions (See new Fig S2D,E). Next, we examined the distribution pattern of the identified motif in all 3'UTRs and generated additional truncated MS2 constructs that either consisted of fragments containing all motifs within a particular 3'UTR or containing reduced numbers (or none) of the motifs (See Fig. S2E). This approach revealed that the motif repeats are highly restricted to 3'UTR sequences that robustly target the reporter mRNAs to cell protrusions (Fig S2E). Finally, we demonstrate that excising a single repeat of the motif within the most proximal region of *TRAK2* 3'UTR fully abolishes the localisation activity of this region. Altogether, our new data show that the motif enriched in *k5* mRNAs has conserved functions across multiple *k5* cluster transcripts. Moreover, given that several proteins encoded by the remaining *k5* mRNAs have also been linked to actin remodelling, it is tempting to speculate that their co-targeting and localised translation in motile cells may participate in a coordinated regulation of actin dynamics at the leading edge. Indeed, this hypothesis will be a focus of future work in the laboratory.

- Supplementary Video 1: it would help to better see the colocalization of the MCP-GFPnls signal and the position of filopodia if the arrowheads pointing to newly formed filopodia were also indicated on the left panel and/or if the position of the MCP-GFPnls signal was indicated in the right panel.

We fully agree with Referee that displaying arrowheads indicating newly-formed filopodia in Supplementary Video 1 would visually strengthen the message that filopodia are generated in close proximity to the cytoplasmic MCP-GFPnls signal. As such, we have added arrowheads to the new version of the video (left panel of Movie S1).

- *RAB13* siRNA knockdown results in increased filopodia formation. Fig3d seems to show more filopodia although quantification in Fig 3e and f indicates decreased number of filopodia

We agree with the Referee that the image presented in Fig. 3D does not identically match quantifications presented in Fig. 3E,F. As discussed in response to Referee 2, the morphology of these cells is highly heterogeneous and the presented examples were primarily selected to emphasise the significant differences in filipodia number observed at distal sites upon loss of *RAB13*. Indeed, it would be very difficult to identify an individual cell that exactly mirrors the output of our average quantified data. Hence, the power of this analyses lies predominantly in the significance of the population level effects of *RAB13* knockdown quantified in Fig. 3F. We have added new text to the manuscript to clarify this point.

- Figure 3b,c: how comes puro-PLA punctae are detected in the -Puro control?

The Referee rightly queries the detection of low levels of Puro-PLA punctae in cell protrusions not exposed to Puromycin (-Puro). In these controls, newly translated protein would not be labelled with Puromycin in the nascent peptide chains and the absence of punctae may be expected. However, with all Puro-PLA analyses there is a low rate of non-specific signal generated by the PLA assay itself, as can be seen in -Puro controls from the original publication reporting this method (tom Dieck *et al.*, 2015), as well as in more recent work investigating *RAB13* translation by Puro-PLA (Moissoglu *et al.*, 2019). Importantly, despite these background counts, there is a clear enrichment of Puro-PLA puncta in protrusions exposed to Puromycin versus the -Puro controls (Fig. 3C).

- There is no reference to Figure 3d in the text

We thank the Referee for indicating the absence of a reference to Fig 3D and we have now added it to manuscript when describing this data on Page 7.

- *rab13d3'UTR/d3'UTR* embryos: why does the miss-directed branching only occur at the horizontal myoseptum and not upon sprouting from the dorsal aorta?

This is an interesting point raised by the Referee and likely relates to the horizontal myoseptum (HM) being a critical decision point during ISV migration. To ensure correct patterning of ISVs, as endothelial tip cells reach the HM they need to need to correctly decide to migrate dorsally past the neural tube, rather than between the neural tube and notochord (See Fig. S7I). As such, there are critical guidance cues (e.g. semaphorin-plexin) that correctly orient migrating endothelial cells. Indeed, previous studies have shown that embryos deficient for such guidance factors display clear branching abnormalities at the HM level (Torres-Vazquez *et al.*, 2004; Lamont, Lamont and Childs, 2009; Lu *et al.*, 2004). Consistent with these observations, Phng *et al.* (2013) demonstrated that sprouting zebrafish ISVs devoid of filopodia excessively pause at HM, further strengthening the hypothesis that filopodia-mediated decision-making drives correct pathfinding during ISV formation at this complex transition point.

Can the aberrant branching phenotype also be observed in other sites of sprouting angiogenesis, like hindbrain vasculature for instance?

The Referee queries whether other vascular beds undergoing angiogenesis display similar sprouting defects to the ISVs upon mutation of *rab13* 3'UTR in zebrafish. To date, we have not detected sprouting defects at other sites. However, as discussed in response to the Referee's comment above, we should note that ISV migration appears to be particularly susceptible to perturbation, likely due to the environmental complexity of this migratory route. Indeed, we observe low rates of defective pathfinding even in wild-type and heterozygous embryos (~3% of embryos versus 40% in *rab13* LE mutants), which we do not observe in other vascular beds. This susceptibility to perturbation is one key reason why the ISVs have proven such a useful tool for defining vascular gene function *in vivo*. As such, we may not expect to observe similar defects in other more developmentally robust vascular beds, unless in examined in a compromised background, (e.g. upon co-depletion of pro-migratory Vegfr-2 signalling).

- What is the effect of induced ISV hyperbranching, e.g. by Notch inhibition, on *rab13* mRNA localization?

The Referee suggests an interesting experiment, but one we were unable to complete before the COVID-19-related closure of research facilities due to technical issues. Firstly, these experiments need to be performed *in vivo*, as such "hyperbranching phenotypes induced by Notch inhibition are not seen in *in vitro* assays and would be lost once zebrafish cells are explanted. But, it is not possible to assess the localisation of *rab13* *in vivo* using smFISH as we cannot distinguish endothelial *rab13* from transcript expressed in adjacent cells. Hence, these studies would require assessment of *rab13* 3'UTR dynamics using the MS2-MCP system combined with Notch perturbation, ideally upon knockdown of the vascular Notch ligand *dll4*. These MCP-GFPnls experiments require embryonic over-expression of exogenous *rab13* 3'UTR-MS2 hairpin fusion constructs, which make embryos exceptionally susceptible to further perturbations and we have not yet been successfully obtaining sufficient

numbers of *rab13* 3'UTR-expressing embryos that we could be subjected to Notch inhibition. However, it is important to note that loss of vascular Notch signalling is well established to invoke an increase in tip cell numbers (Hellström *et al.*, 2007; Lobov *et al.*, 2007; Siekmann and Lawson, 2007; Suchting *et al.*, 2007). As such, we expect more cells to exhibit tip cell-like patterning of *rab13* mRNA to the front of the cell, but we would not expect any particular shift in the localisation of *rab13* mRNA, as these extra cells will behave just like the tip cells we have already characterised in Fig. 4A and S6B (Movies S2 and S3). In particular, whilst being a potentially interesting confirmation of Notch function during vascular development, the results of this experiment would not shift the core message of the manuscript regarding the role played by *rab13* mRNA localisation in the spatial control of motile cell polarity and tissue formation.

- Supplementary Video 2: it is hard to distinguish the MCP-GFPnls signal in this line and to look at the distribution of the signal over the cell. Why is the localization of the *rab13* mRNA different than *in vitro* (in the filopodia rather than at the base)? Despite what is mentioned in the text, the cell in Video 2 is not a tip cell. It would be better to look at the *rab13* mRNA distribution in true tip cell

The Referee raises several points related to the MS2-MCP *in vivo* mRNA labelling presented in Fig. 4A. Firstly, to aid interpretation of the localisation of the *rab13* LE *in vivo*, we have additional data and movies of both a migrating endothelial tip cell (Fig. S6B, Movie S3) and stalk cell (Fig. S6C, Movie S4). Both these movies confirm that expression of the *rab13* LE drives MCP-GFPnls targeting to distal sites of migrating cells. Moreover, these movies highlight how *rab13* mRNA can be located both within filopodia (Fig. 4A) and at the base of filopodial extensions (Fig. S6B, Movie S3). As such, the localisation of *rab13* mRNA is not “different” to cells *in vitro*, but occasionally exhibits a more prominent localisation within filopodia extensions likely due to the greater size of these structures *in vivo*.

- The region that was excised from the 3'UTR of the zebrafish *rab13* transcript is quite big compared to the minimal localization element determined in HUVEC, this might affect also other regulatory mechanisms. Can the same RNA motifs and minimal LE be determined in the zebrafish 3'UTR region allowing more precise deletion?

We agree with the Referee that comprehensive dissection of the zebrafish 3'UTR would be preferable. However, it is important to point out that mutation of the zebrafish *rab13* LE did not influence mRNA expression/stability (Fig. S7J) and does not perturb protein production, as the observed phenotype does not resemble that seen upon Rab13 knockdown using CRISPRi (Fig. III above) or MO tools (Wu *et al.*, 2011). Hence, we are highly confident that the observed phenotype is not a consequence of defects in “other regulatory mechanisms”. Nevertheless, we agree that it would be ideal to identify and mutate a minimal LE within the zebrafish 3'UTR, but this is not possible for a number of reasons. First, as discussed in response to other Referees' comments, due to the large evolutionary distance between the human and zebrafish genomes, there is low sequence conservation between the *RAB13/rab13* LE sequence, despite high conservation of the function of this 3'UTR region. Hence, identification of a homologous zebrafish localisation motif has proven difficult without a substantial zebrafish-specific repeat of the multi-tissue RNAseq studies, clustering analyses and motif searches performed in human cells in Fig 1/S1. Moreover, we were unable to use heterologous studies in human cells to rapidly dissect the zebrafish 3'UTR due to this lack of conservation between species. Finally, there are limited options for gRNAs specifically targeting sites within the *rab13* 3'UTR that make the generation of multiple zebrafish mutant lines with distinct 3'UTR deletions impossible. But, importantly, we prove that *rab13* polarisation (but not expression) is elegantly disturbed in our *rab13* 3'UTR mutant embryos, indicating that this region does indeed encompass the functional LE and is a highly appropriate tool for testing the function of *rab13* mRNA targeting *in vivo*.

- What is the trans-acting partner to mediate targeted localization of Rab13 mRNA and likely also the other cluster k5 members?

We apologise for a lack of clarity regarding published work on the mechanisms of *RAB13* mRNA transport, as it is well established that *RAB13* and cluster *k5* mRNAs are dependent on the activity of APC, a plus-end microtubule associated protein implicated in the localisation of mRNAs to cell protrusions (Mili, Moissoglu and Macara, 2008; Wang *et al.*, 2017).

- It would be interesting to also find out the functional role of the other mRNAs with conserved polarization

We agree with the Referee that the function of other cluster *k5* mRNA is of great interest, and in particular, if they function in parallel with *RAB13* to modulate cell polarity and tissue movement. Indeed, many of these mRNAs share known roles in actin remodelling, suggesting complementary functions to *RAB13* in the modulation of local process formation/retraction. In new data, we have truncated the 3'UTRs of all cluster *k5* mRNA to define the key motif/s responsible for mRNA targeting (Fig. S2D,E). Hence, we are now poised to conduct such suggested functional studies of *KIF1C*, *NET1*, *RASSF3*, *TRAK2* mRNA targeting *in vitro*. However, this work will still require a huge effort to generate/phenotype HUVEC CRISPR mutants, identify orthologous LEs in zebrafish and

generate/phenotype zebrafish CRISPR mutants for these genes. As such, there are years of work ahead of us before we can describe equivalent roles to other *k5* mRNAs in the modulation of motile cell polarity and tissue morphogenesis. Hence, we feel this work is beyond the scope of the current submission and itself will constitute several independent manuscripts.

- Is the effect of the LE deletion a real loss of function or rather a gain of function that causes “misguidance”. What happens when overexpressing the deletion mutant in the background of WT Rab13mRNA? This may help to distinguish between the effects.

The Referee suggests an experimental procedure to test whether disruption of *rab13* mRNA targeting results in loss or gain of function of Rab13 activity. However, it is important to note that we do not believe this phenotype is a consequence of either a loss or gain of function, but arises from a shift in the *location* of Rab13 activity. Indeed, this question was elegantly addressed in a parallel study by Moissoglu et. al. that is also under consideration at *EMBO Journal*. In this work, Moissoglu and colleagues demonstrate that disruption of *RAB13* mRNA polarisation using morpholino oligonucleotides against the LE results in a shift in location of activated RAB13. Moissoglu et. al. show that perturbation of *RAB13* mRNA targeting to protrusions blocked the co-translational interaction of RAB13 with its GEF (RABIF) at this peripheral site, but not at other locations within the cell. Hence, phenotypes observed upon loss of *RAB13* mRNA polarisation are not due to a global gain or loss of function, but to a switch in the site of RAB13 activity. As such, the observations of Moissoglu et. al. perfectly complements our work by providing a solid mechanistic basis for why a change in the location of *RAB13* targeting can broadly disrupt motile cell polarity and oriented tissue movement - i.e. via displacement of the normal site of RAB13 activity. In light of this new data, interpretation of results from the experimental procedure suggested by the Referee would be very difficult considering the phenotype is not simply a consequence of Rab13 loss or gain of function. Moreover, as the *rab13* LE mutant should retain full functionality at non-protrusion intracellular sites, global overexpression of this mRNA would likely induce early developmental defects that may preclude analysis of later vessel phenotypes.

References

- El-Brolosy, M. A., Kontarakis, Z., Rossi, A., Kuenne, C., Günther, S., Fukuda, N., Kikhi, K., Boezio, G. L. M., Takacs, C. M., Lai, S. L., Fukuda, R., Gerri, C., Giraldez, A. J. and Stainier, D. Y. R. (2019) 'Genetic compensation triggered by mutant mRNA degradation', *Nature*, 568(7751), pp. 193-197.
- Farnsworth, D. R., Saunders, L. M. and Miller, A. C. (2020) 'A single-cell transcriptome atlas for zebrafish development', *Dev Biol*, 459(2), pp. 100-108.
- Harris, E. S. and Nelson, W. J. (2010) 'Adenomatous polyposis coli regulates endothelial cell migration independent of roles in beta-catenin signaling and cell-cell adhesion', *Mol Biol Cell*, 21(15), pp. 2611-23.
- Hellström, M., Phng, L. K., Hofmann, J. J., Wallgard, E., Coultas, L., Lindblom, P., Alva, J., Nilsson, A. K., Karlsson, L., Gaiano, N., Yoon, K., Rossant, J., Iruela-Arispe, M. L., Kalén, M., Gerhardt, H. and Betsholtz, C. (2007) 'Dll4 signalling through Notch1 regulates formation of tip cells during angiogenesis', *Nature*, 445(7129), pp. 776-80.
- Hurlstone, A. F., Haramis, A. P., Wienholds, E., Begthel, H., Korving, J., Van Eeden, F., Cuppen, E., Zivkovic, D., Plasterk, R. H. and Clevers, H. (2003) 'The Wnt/beta-catenin pathway regulates cardiac valve formation', *Nature*, 425(6958), pp. 633-7.
- Ioannou, M. S., Bell, E. S., Girard, M., Chaineau, M., Hamlin, J. N., Daubaras, M., Monast, A., Park, M., Hodgson, L. and McPherson, P. S. (2015) 'DENND2B activates Rab13 at the leading edge of migrating cells and promotes metastatic behavior', *J Cell Biol*, 208(5), pp. 629-48.
- Kharel, P., Balaratnam, S., Beals, N. and Basu, S. (2020) 'The role of RNA G-quadruplexes in human diseases and therapeutic strategies', *Wiley Interdiscip Rev RNA*, 11(1), pp. e1568.
- Lamont, R. E., Lamont, E. J. and Childs, S. J. (2009) 'Antagonistic interactions among Plexins regulate the timing of intersegmental vessel formation', *Dev Biol*, 331(2), pp. 199-209.
- Lobov, I. B., Renard, R. A., Papadopoulos, N., Gale, N. W., Thurston, G., Yancopoulos, G. D. and Wiegand, S. J. (2007) 'Delta-like ligand 4 (Dll4) is induced by VEGF as a negative regulator of angiogenic sprouting', *Proc Natl Acad Sci U S A*, 104(9), pp. 3219-24.
- Lu, X., Le Noble, F., Yuan, L., Jiang, Q., De Lafarge, B., Sugiyama, D., Breant, C., Claes, F., De Smet, F., Thomas, J. L., Autiero, M., Carmeliet, P., Tessier-Lavigne, M. and Eichmann, A. (2004) 'The netrin receptor UNC5B mediates guidance events controlling morphogenesis of the vascular system', *Nature*, 432(7014), pp. 179-86.
- Marzesco, A., Dunia, I., Pandjaitan, R., Recouvreur, M., Dauzonne, D., Benedetti, E. L., Louvard, D., Zahraoui, A. (2002) 'The Small GTPase Rab13 Regulates Assembly of Functional Tight Junctions in Epithelial Cells', *Mol Biol Cell*, 13(6), pp. 1819-31.
- Mili, S., Moissoglu, K. and Macara, I. G. (2008) 'Genome-wide screen reveals APC-associated RNAs enriched in cell protrusions', *Nature*, 453(7191), pp. 115-9.
- Moissoglu, K., Yasuda, K., Wang, T., Chrisafis, G. and Mili, S. (2019) 'Translational regulation of protrusion-localized RNAs involves silencing and clustering after transport', *Elife*, 8.
- Nokes, R. L., Fields, I. C., Collins, R. N. and Fölsch, H. (2008) 'Rab13 regulates membrane trafficking between TGN and recycling endosomes in polarized epithelial cells', *J Cell Biol*, 182(5), pp. 845-53.
- Park, H. Y., Trcek, T., Wells, A. L., Chao, J. A. and Singer, R. H. (2012) 'An unbiased analysis method to quantify mRNA localization reveals its correlation with cell motility', *Cell Rep*, 1(2), pp. 179-84.
- Phng, L. K., Stanchi, F. and Gerhardt, H. (2013) 'Filopodia are dispensable for endothelial tip cell guidance', *Development*, 140(19), pp. 4031-40.
- Rossi, A., Kontarakis, Z., Gerri, C., Nolte, H., Hölper, S., Krüger, M. and Stainier, D. Y. (2015) 'Genetic compensation induced by deleterious mutations but not gene knockdowns', *Nature*, 524(7564), pp. 230-3.

Siekmann, A. F. and Lawson, N. D. (2007) 'Notch signalling limits angiogenic cell behaviour in developing zebrafish arteries', *Nature*, 445(7129), pp. 781-4.

Subramanian, M., Rage, F., Tabet, R., Flatter, E., Mandel, J. L. and Moine, H. (2011) 'G-quadruplex RNA structure as a signal for neurite mRNA targeting', *EMBO Rep*, 12(7), pp. 697-704.

Suchting, S., Freitas, C., le Noble, F., Benedito, R., Bréant, C., Duarte, A. and Eichmann, A. (2007) 'The Notch ligand Delta-like 4 negatively regulates endothelial tip cell formation and vessel branching', *Proc Natl Acad Sci U S A*, 104(9), pp. 3225-30.

tom Dieck, S., Kochen, L., Hanus, C., Heumuller, M., Bartnik, I., Nassim-Assir, B., Merk, K., Mosler, T., Garg, S., Bunse, S., Tirrell, D. A. and Schuman, E. M. (2015) 'Direct visualization of newly synthesized target proteins in situ', *Nat Methods*, 12(5), pp. 411-4.

Torres-Vazquez, J., Gitler, A. D., Fraser, S. D., Berk, J. D., Van, N. P., Fishman, M. C., Childs, S., Epstein, J. A. and Weinstein, B. M. (2004) 'Semaphorin-plexin signaling guides patterning of the developing vasculature', *Dev Cell*, 7(1), pp. 117-23.

Wang, T., Hamilla, S., Cam, M., Aranda-Espinoza, H. and Mili, S. (2017) 'Extracellular matrix stiffness and cell contractility control RNA localization to promote cell migration', *Nat Commun*, 8(1), pp. 896.

Wu, C., Agrawal, S., Vasanji, A., Drazba, J., Sarkaria, S., Xie, J., Welch, C. M., Liu, M., Anand-Apte, B. and Horowitz, A. (2011) 'Rab13-dependent trafficking of RhoA is required for directional migration and angiogenesis', *J Biol Chem*, 286(26), pp. 23511-20.

Zhang, L., Dai, F., Cui, L., Zhou, B. and Gio, Y. (2017) 'Up-regulation of the active form of small GTPase Rab13 promotes macroautophagy in vascular endothelial cells', *Biochim et Biophys Acta Mol Cell Res*, 1864(4), pp. 613-24.

Thank you for submitting your manuscript entitled "mRNA compartmentalisation spatially orients tissue morphogenesis" [EMBOJ-2020-106003] to The EMBO Journal. I have discussed it and the existing referees' reports with the editorial team. In addition, the study and the point-by-point rebuttal letter have been sent to one referee for evaluation.

This reviewer concurs with us on the general interest of your findings and supports publication of the study in The EMBO Journal. In addition, (s)he gives you some suggestions on how to streamline and improve the clarity of the manuscript.

Given the interest in the study and the positive recommendation from a trusted expert in the field, we are offering to pursue publication of your manuscript, pending text reformatting, extension of the discussion section, and clarification of some editorial issues.

REFEREE REPORTS

Referee #1:

I agree that the authors have addressed the reviewers' comments satisfactorily. In principle the work is well suited for EMBO J, especially given the complementary, mechanistic findings in the Mili study. However, I recommend that the authors take advantage of the longer word limit at EMBO J to explain the new additions, as well as previous experiments and controls, in more detail (several important results in the supplement are skimmed over). This will improve the clarity and accessibility of the manuscript. For example, controls for off-target effects should be addressed directly in the text, as should the observation of consistent phenotypes in independent clones. There are other concerns of the reviewers (which may be shared by readers that have not had time to scrutinise the Methods or Legends) that the authors may wish to address directly in the text. Furthermore, the findings of the Mili lab study and how they complement this study should also be discussed in the final manuscript. Any key discrepancies between the two studies could also be discussed.

I would like to see some discussion of the potential discrepancy between APC functions and Rab13 LE functions (see rebuttal letter) but I will lead it to the authors to decide what is appropriate.

There are still several grammatical errors and typos, so careful editing is required.

Referee #1 Comments:

I agree that the authors have addressed the reviewers' comments satisfactorily. In principle the work is well suited for EMBO J, especially given the complementary, mechanistic findings in the Mili study. However, I recommend that the authors take advantage of the longer word limit at EMBO J to explain the new additions, as well as previous experiments and controls, in more detail (several important results in the supplement are skimmed over). This will improve the clarity and accessibility of the manuscript. For example, controls for off-target effects should be addressed directly in the text, as should the observation of consistent phenotypes in independent clones. There are other concerns of the reviewers (which may be shared by readers that have not had time to scrutinise the Methods or Legends) that the authors may wish to address directly in the text.

As suggested by the Referee, we have now restructured the Figure panels and Results text to fully explain all key control experiments and new data generated during manuscript revision. In particular, we have now (1) fully discussed our control RNAseq screens for off-target effects in the main Results text, (2) moved data confirming consistent phenotypes in independent clones from supplemental data to the main Figure panels, and (3) moved all control data related to CRISPR-Cas9 experiments from supplemental data to the main Figure panels.

Furthermore, the findings of the Mili lab study and how they complement this study should also be discussed in the final manuscript. Any key discrepancies between the two studies could also be discussed.

As requested, we have now included a full discussion of the back-to-back work by Mili and coworkers in the Discussion text.

I would like to see some discussion of the potential discrepancy between APC functions and Rab13 LE functions (see rebuttal letter) but I will lead it to the authors to decide what is appropriate.

We agree with the Referee that this is a key point, hence, we have added a full discussion of the discrepancy between APC and *RAB13* LE function to the Discussion text.

There are still several grammatical errors and typos, so careful editing is required.

As requested, we have now addressed these grammatical errors.

Thank you for submitting your revised manuscript. I have now checked it and found few minor issues that need to be addressed before we can officially accept your study.

2nd Authors' Response to Reviewers**8th Aug 2020**

The Authors' have made all the requested editorial changes

Accepted**14th Aug 2020**

I am pleased to inform you that your manuscript has been accepted for publication in The EMBO Journal.

Corresponding Author Name: Guilherme Costa and Shane P. Herbert
Journal Submitted to: EMBO Journal
Manuscript Number: 106003